# The use of humanure for cereal production under conventional and regenerative farming models - findings from a three-year grassland-to-arable transition

**Katie Allen**[1]*, **Effie Papargyropoulou**[2], **Ruth Wade**[3], **Barbara Evans**[1]

**1** School of Civil Engineering, University of Leeds, United Kingdom Of Great Britain and Northern Ireland, **2** School of Earth and Environment, University of Leeds, United Kingdom Of Great Britain and Northern Ireland, **3** School of Biology, University of Leeds, United Kingdom Of Great Britain and Northern Ireland

* cnkma@leeds.ac.uk

## Abstract

Humanure (human excrement recycled for agricultural use) offers a low-cost, simple treatment option for dry sanitation systems, and a promising organic amendment for crop production. However, no long-term studies have evaluated its impacts under different land management approaches. This study presents findings from a three-year field trial (2021–2024) in West Yorkshire, UK, assessing the effects of humanure on cereal crop yield and soil properties under conventional and regenerative farming practices of a former grassland. The experiment used a semi-randomised block design with two management regimes (conventional and regenerative) and three fertilisation treatments (control, synthetic fertiliser, and humanure), each replicated three times. Crop results showed that humanure increased crop yields compared to the control, demonstrating a fertilisation effect, which was more pronounced under regenerative management. However, yields from humanure treatments were lower than those from synthetic fertilisers. Regenerative management produced higher yields than conventional across all fertiliser types. Soil analyses revealed that fertiliser type had limited influence on physical soil properties, but regenerative practices significantly improved water-stable aggregates. Soil organic matter and total carbon were higher under humanure than other fertiliser regimes, and higher under regenerative management across all fertilisers. Humanure elevated phosphorus and potassium levels, but not outside of acceptable limits. Biological indicators, including worm counts, soil respiration, and fungal biomass, were slightly elevated under humanure treatments and consistently higher in regenerative systems. Due to small sample sizes, detailed statistical analysis was not appropriate, and instead trends are observed and discussed. Overall, this study suggests that humanure is a promising soil amendment, especially when combined with regenerative land management. More investigation is warranted, with greater replication to enable statistical analysis,

**Data availability statement:** All relevant data are within the paper and its Supporting Information files.

**Funding:** The author(s) received no specific funding for this work.

**Competing interests:** The authors have declared that no competing interests exist.

in order to further explore of the role humanure in realising a circular economy for sanitation and sustainable agriculture.

## Introduction

It is estimated that global human fecal biomass production will exceed 1 billion tonnes by 2030 [1]. Yet currently only 57% of sanitation systems worldwide are safely managed [2], representing not only a public health crisis, but also a missed opportunity to capture and recycle the valuable nutrients and organic matter within fecal material.

At the same time, population growth, climate change, and changing consumer demands are placing increasing pressure on global food production systems, prompting further exploration into strategies for food system transformation [3]. Numerous solutions have been proposed, both for reducing losses and increasing supply. However, approaches to increase supply, principally characterised as either agricultural expansion or intensification, are often criticised for their environmental consequences [4,5], underscoring the need for strategies for more sustainable food production systems. One such criticism, relevant to the present study, is the reliance of intensive arable systems on synthetic chemical fertiliser inputs, which contribute significantly to greenhouse gas emissions [6] and create widespread economic and social vulnerabilities due to dependence on volatile global supply chains [7,8].

Together, these drivers present a compelling case for exploring the use of human fecal material as a source of nutrients and organic matter in agricultural production. The abundance, locality, affordability and nutrient content of fecal material make this a promising option for both sanitation management and as part of a more sustainable food system, closing the nutrient-loop between sanitation and food production.

### Humanure

Previous research on the agricultural use of human fecal material is concentrated on wastewater and biosolids; the liquid and semi-solid outputs of sewer-based wastewater treatment plants. Comparatively little attention is given to alternative toilet systems, which contain the material in situ and require periodic emptying [9]. Despite representing the most common form of sanitation service globally [2], these systems have been largely overlooked in the context of agricultural reuse.

Dry toilets which do not use flushwater are one such system, and require different management to sewage. One promising treatment option which is suitable for agricultural reuse is composting, due to its simplicity, low cost, and broad applicability [10]. In contrast to extractive technologies (e.g., phosphorus extraction [11] composting utilises the entirety of the toilet material, aligning it with zero waste and circular economy principles [12,13]. Composted human feces are henceforth referred to as 'humanure', a term popularised by Jenkins (1994) [14].

Previous studies on the land application of humanure have reported improved crop growth relative to unfertilised controls, as well as increases in soil organic matter (SOM), nitrogen, and phosphorus levels—indicating a promising fertilisation effect

[15–17]. However, a recent systematic review of fecal amendments in agricultural trials highlighted a critical lack of multi-year field studies and insufficient reporting on key soil health indicators, particularly those related to physical and biological properties [9]. This limits our understanding of the longer-term and systemic impacts of humanure on soil function.

Moreover, existing research on feces-based amendments has typically been conducted without situating trials within clearly defined or deliberately modelled farming systems. As a result, we lack understanding of how the performance of such amendments differs under varying land management approaches. With growing recognition of the need for food production transformation, it is important that research into alternative fertilisers considers their performance within different farming contexts, in order to assess its real-world viability, and to identify those contexts in which it can be most effective.

## Grassland-to-arable transition

The conversion of grassland to arable land is relevant to the discussion of food system transformation in several ways. Whether through the conversion of marginal lands into food production, or the transition of pasture lands into arable production, several drivers may underpin this practice in different contexts. Climate change is altering the geographic suitability of different forms of food production, and may make new areas suitable for arable production [18]. Evolving consumer preferences in some regions, such as reduced meat consumption and greater demand for plant-based foods, may create pressure to shift land use away from pasture and into arable production [19,20]. And finally, expansion of arable farming into marginal or unused sites is cited as a strategy to increase food production capacity, especially in urban areas [21]. When considering urban areas, it should also be noted that urban centres produce the highest density of fecal material, and so the conversion of urban and peri-urban spaces into arable food production could facilitate closer linkages between supply and demand for nutrients [22].

The present study models this land-use transition in practice, converting grassland to arable land across three years. Within this framework, two contrasting management approaches were examined, termed "conventional" and "regenerative". Terms used to describe agricultural systems are often oversimplified or politicised, which can obscure the immense diversity of farming practices [23]. As recommended by Newton et al. (2020) [24] a detailed list of the specific "conventional" and "regenerative" practices used within this study is included in the methods, and these terms are used as a tool to simplify the discussion. The authors acknowledge the complexity of farming systems and understand that the results presented here do not represent all systems globally. Rather, it is hoped that this exploratory study will highlight interesting trends and inform future research in the field.

## Study rationale

The present study was a field-plot trial which examined the effects of humanure on cereal crop growth and soil health over a three-year grassland-to-arable transition. It compared the performance of humanure to recommended synthetic fertiliser dose and to no fertilisation, modelled under either a "conventional" or "regenerative" suite of land management practices. This broadens the focus from simply fertiliser input-substitution, an approach which is commonplace in agricultural research and criticised for failing to challenge the damaging, input-dependant structure of industrialised food production, and thus limiting the potential to achieve truly sustainable agriculture [25].

This experiment followed trial design, data collection and reporting recommendations outlined in Allen et al. [9].

## Conventional management experiment

What is considered "conventional" varies globally, making cross-study comparison and broad generalisation challenging. Typically, conventional agriculture refers to large-scale, mechanised farming operations with minimal crop diversity (e.g., monocultures) and a high reliance on external inputs of crop nutrients and other agrichemicals. This study models a typical UK cereal production system, incorporating mechanical tillage and pesticide use. More details are presented in the Methods section.

The primary objectives of this system within the study are to offer comparison against typical 'best practice' for UK cereal production, and to evaluate the direct substitution of synthetic fertiliser with humanure within such a system.

Synthetic fertilisers are designed for use in conventional, intensive farming systems, where they supply highly soluble forms of nitrogen, phosphorus, and potassium (NPK) that are immediately available for plant uptake. In such systems, soil biological processes that typically support nutrient cycling are often disrupted—due to factors like intensive tillage and chemical pest control—which impairs the soil's natural ability to supply endogenous nutrients to plants [26–29]. Synthetic fertilisers effectively bypass the need for a healthy, functioning soil ecosystem by directly delivering nutrients, making them an attractive solution in degraded or biologically simplified systems.

In contrast, humanure contains large quantities of these elements in organic forms, which require mineralisation by soil microorganisms before they can become available to plants, resulting in a slower release of plant-available nutrients [30,31].

Therefore, it is theorised that direct input substitution of synthetic fertiliser for humanure would see an initial reduction in crop yield, but that this yield gap may close over multiple seasons as the nutrients become available. Additionally, it is theorised that, due to its organic matter content, humanure will offer benefits for soil health which are not observed under synthetic fertilisation [15,17,32–37].

The review by Allen et al. (2023) [9] found only four high quality studies to date which had directly compared the effects of humanure and synthetic fertiliser on crop growth [15,38–40], all of which took place over a single cropping cycle of a few months, highlighting the need for more long-term research.

### Regenerative management experiment

"Regenerative" agriculture also encompasses a wide range of practices [41]. Broadly, it aims to restore soil and ecosystem health through methods including minimising soil disturbance, maintaining living roots year-round, and increasing plant diversity. In this study, the regenerative system includes minimal tillage, no agrichemical inputs, cover cropping, and a ley year without a market crop (full details in the Methods section).

The objective of this system within the study is to assess how humanure might contribute to soil regeneration, and how it may offer greater fertility benefits within a regeneratively managed system as opposed to a conventional one.

Since the organically-bound nutrients contained within organic materials require mineralisation to become plant-available [30], it is theorised that the fertilising capability of humanure will depend not only on its nutrient content, but also on the biological activity and overall health of the surrounding soil system. Given this, humanure is expected to deliver comparatively greater crop fertilisation effects under systems which focus on regeneration of soil ecosystems and promotion of biological soil health.

### Research questions and hypotheses

The study rationale raises three research questions:

- **RQ1: How does humanure application affect cereal crop growth compared to synthetic fertiliser and no fertilisation?**

- **RQ2: How does humanure influence soil health parameters during grassland-to-arable transition?**

- **RQ3: Do these effects differ between "conventional" and "regenerative" management systems?**

These questions are addressed through the following hypotheses:

- **H1: Humanure will enhance cereal crop growth relative to no fertilisation.**

- **H2: Humanure will initially produce lower yields than synthetic fertiliser due to slower nutrient mineralisation, but will approach parity over successive seasons.**

° **H2a: The fertilising effect of humanure will be more pronounced under regenerative management practices, where enhanced soil biological activity supports nutrient mineralisation and cycling.**

- **H3: Humanure will help to mitigate declines in soil health during the grassland-to-arable transition by supplying organic matter which supports soil physical properties (e.g., bulk density and water retention), and enhances biological activity.**

° **H3a: These soil health benefits will be greater under regenerative management, where reduced disturbance limits losses of organic matter and microbial biomass.**

## Materials and methods

### Experiment overview

A plot trial was conducted over three years with three crop growth cycles, from June 2021 to September 2024.

It followed a 3x2 factorial design, where six treatment/management combinations were applied across the study site, each replicated three times. The three fertiliser treatments were a control (no fertiliser), humanure and synthetic fertiliser, which were applied based on a matched total nitrogen loading rate. The two land management types were conventional and regenerative practices.

This approach allowed for the assessment of both individual effects of fertiliser type and land management, as well as potential interactions between these factors.

Multiple parameters of crop growth and soil health were measured following the third cropping cycle. This study reports the crop and soil results from the end of the third cropping cycle, in September 2024. Some baseline data (2021) and year two data (2023) are included for some soil parameters to identify changes over time.

### Site details

The plots were located on the University of Leeds research farm, located in Tadcaster, West Yorkshire (53°52'00.0"N 1°19'51.0"W). The site is owned by the university, and so no additional permits were required to conduct the research. The study site covered an area of 0.1 ha which had previously been used for grass growing trials. The soil is a clay loam texture.

The plots are located within a Nitrate Vulnerable Zone, which placed restrictions on the timing and quantity of Nitrogen applications from different sources.

### Plot layout

The research area measured 48 x 25m and was divided into 24 plots of 1.5m width and 12m length (Fig 1). The layout was semi-randomised block design, comprising three blocks (A, B and C). Each block contained one regenerative management row and one conventional management row. This design enabled access to the entire row for easier use of machinery. Each row contained one of each fertiliser treatment and a discard plot, with positions within the row randomised. Fertilisation was performed by hand.

### Baseline soil assessment

Prior to the start of the experiment, the research area was treated with herbicide (Motif, 5L/ha) in June and August 2021, before the entire area was power harrowed twice in September. This terminated weeds and prepped a uniform bed to begin the trial.

Baseline soil testing was conducted in September 2021, the day before the first crops were planted. The sampling protocol was adapted from the Cornell Framework [42]. Ten samples were taken from around the site in two W shapes and mixed to produce one composite sample. The sampling depth was 15 cm.

**Fig 1. Experimental plot layout and plot codes.** Vertical rows alternate between conventional (pink) and regenerative (green) management, separated by a 2m gap. Each row included one of each of the treatments in a randomised position; control (yellow), humanure (brown), synthetic fertiliser (blue) and a discard plot (grey).

## Land and crop management

The trial compared two different management practices; termed "conventional" and "regenerative".

Table 1 highlights the key management differences between the conventional and regenerative plots. The conventional management approach was modelled on the AHDB growth guides for wheat and barley [43,44]. A more detailed list of plot management activities is included in the Supporting Information (S1 Table).

The crops and varieties selected were typical for a UK cropping rotation, consisting of winter wheat, followed by two years of spring barley.

## Fertiliser applications

The trial compared the impacts of three different sources of crop nutrition. These are:

• Control (Con) – No fertiliser application

• Synthetic fertiliser (SF) – Commercial solid prilled NPK fertiliser ('Origin'), 20-10-10 compound blended mix. These contain (by weight) 20% total N, 10% phosphate ($P_2O_5$) and 10% potash ($K_2O$).

• Humanure – Human feces, urine, toilet paper and sawdust removed from composting-style toilets, stored and aged for at least 2 years to achieve acceptable pathogen die-off for reuse.

The humanure and SF were surface applied by hand during periods of dry weather.

The SF was applied at the recommended nitrogen loading rate and timings, from the AHDB crop growth guides [43,44] and consultation with the site trial manager.

Each humanure batch was sourced from different locations around the UK, meaning that its composition differed each year. A summary of the batch collection and management, as well as a full compositional analysis, is given in the Supporting Information (S1 Appendix and S2 Table).

The nitrogen composition of the humanure was used to calculate the quantity of humanure to be applied to the plots each year, in order to closely match the total nitrogen ($N_{tot}$) loading rate between the humanure and synthetic fertilisers

**Table 1. Overview of key management activities undertaken on the conventional and regenerative plots. Full details are given in S1 Table.**

| | Conventional | Regenerative |
|---|---|---|
| Ground preparation | Deep plough (8 inches) each year to terminate weeds and bury crop residue. Surface power harrowed each year to prepare seedbed for planting. | Surface power harrowed prior to Year 3 only, in order to penetrate thick crop residue. |
| Sowing | Direct drill and rolled | Direct drill and rolled |
| Crops | Year 1 – Winter wheat (*Syngenta Gleam*) Year 2 – Spring barley (*LG Diablo*) Year 3 – Spring Barley (*Laureate*) | Year 1 – Winter wheat (*Syngenta Gleam*) + red clover (*Avisto*) understory Year 2 – Living mulch of red clover (*Avisto*) and fodder radish. Year 3 – Spring Barley (*Laureate*) |
| Chemical Sprays | Pre-emergence herbicide, and fungicide application as recommended in the AHDB growth guides. Spot use of glyphosate for weed control as required. | None |
| Winter cover | Fallow | Cover crop of red clover, mowed off prior to Y3 sow. |

(Table 2). For practicality, this was done by calculating the number of kilograms of humanure required to be added to each plot, based on the $N_{tot}$ concentration and moisture content.

The uncertainty around the mean applied values is expressed in Table 2 as upper and lower bounds, shown in brackets. This uncertainty comes from the variance in $N_{tot}$ content, moisture content, and error associated with weighing.

In addition to $N_{tot}$, Table 2 also shows the calculated doses of total nitrogen ($N_{tot}$), inorganic nitrogen ($N_{in}$), organic matter (OM), carbon ($C_{tot}$), phosphorus and potassium which were applied to each plot each year. For synthetic fertiliser, the quantities of total elemental phosphorus ($P_{tot}$) and total potassium ($K_{tot}$) were calculated from the dose of NPK 20:10:10 applied, which is 10% phosphate ($P_2O_5$) and 10% potash ($K_2O$) by mass. For humanure it was not possible to calculate $P_{tot}$ and $K_{tot.}$ Instead available P ($P_{av}$) and exchangeable potassium cations (K+) were measured in the labs. The same lab methods used for soil analysis given in the Methods section were also used for the humanure batch analysis.

### Sampling and laboratory tests

**Soil sampling and laboratory work.** Soil samples were taken after harvest, on 21st August 2023 (Year 2) and 3rd October 2024 (Year 3). Five samples were taken from each plot using the same Cornell method as used for baseline soil analysis [42], to a depth of 15 cm. These were combined to create one composite sample for each plot. Composite samples were sieved to 8 mm to remove large stones and roots, then half of the material was stored fresh at 4°C, whilst the other was air dried.

Bulk density samples were collected using a soil corer with a cutting shoe. The internal cylindrical rings were 5 cm diameter and 5 cm height (volume 98.17 $cm^3$). These were collected at a depth of 5–10 cm, by removing the first 5 cm of topsoil before sampling.

Lab analyses were conducted to determine key physical, chemical and biological properties of the soil. Table 3 summarises the methods used. Full laboratory methods are shown in the Supporting Information (S2 Appendix).

**Table 2. Estimated quantity of Nitrogen, and Organic Matter, Phosphorus and Potassium added to each of the plots each year. Numbers in brackets indicate the maximum and minimum possible values based on the error.**

| | | Year 1* | | Year 2 | | Year 3 | |
|---|---|---|---|---|---|---|---|
| | | Humanure 1 | SF | Humanure 2 | SF 2 | Humanure 3 | SF 3 |
| *Recommended dose and timing $N_{tot}$ (kgN/ha)* | | *Early Spring −100 Mid Spring − 100 Late Spring − 50* | | *Late Spring − 140* | | *Mid Spring − 65 Late Spring − 65* | |
| $N_{tot}$ | kg/ha | **232 [181, 339]** | **250** | **156 [139, 169]** | **140** | **198 [180, 268]** | **130** |
| $N_{In}$ | kg/ha | 48.23 [16.65, 78.56] | 250 | 32.96 [26.54, 40.16] | 140 | 20.19 [16.15, 57.49] | 130 |
| OM | t/ha | 6.98 [5.45, 9.37] | – | 6.19 [5.62, 6.80] | – | 6.43 [5.73, 8.69] | – |
| $C_{tot}$ | t/ha | 3.15 [2.69, 4.07] | – | 2.45 [2.24, 2.66] | – | 3.05 [2.77, 4.11] | – |
| $P_{av}$ | kg/ha | 1.95 [1.61, 2.65] | – | 16.78 [13.24, 20.77] | – | 10.88 [5.94, 15.78] | – |
| $P_{tot}$ | kg/ha | – | 54.55 | – | 30.55 | | 28.37 |
| K+ | kg/ha | 36.21 [7.12, 82.41] | – | 130.79 [110.68, 153.09] | – | no data | – |
| $K_{tot}$ | kg/ha | – | 103.75 | – | 58.1 | – | 53.95 |

*An additional 50 kg/ha of $N_{in}$ was erroneously applied to all plots in the first year, including the control plots, in the form of 20:10:10 NPK Synthetic Fertiliser (262g).

**Table 3. Summary of measured soil parameters and laboratory methods used.**

| Parameter | Summary of Method | References |
|---|---|---|
| Bulk density | Oven dried at 105°C for 12 hours then weighed. | [45] |
| Water Stable Aggregates | Air dry soil sieved to 2 mm, placed onto a 1 mm sieve and submerged in water for 5 mins, then removed and submerged again 5 times to dislodge slaked soil, before removal, drying and weighing to determine proportion retained on the sieve. Indicates the stable fraction of particles sized between 1 mm and 2 mm. | [46] |
| pH | Analysed using Oakton® pH 700 Benchtop pH Meter | [47] |
| Moisture | Oven dried at 105°C for 12hrs, then weighed. | |
| Soil Organic Matter (SOM) | Loss on ignition.<br>Heated to 550°C for 12 hours, then weighed. | [48] |
| Total carbon ($C_{tot}$) and total nitrogen ($N_{tot}$) | Air-dried soil ground on a Resch mixer mill MM400.<br>Analysed using an Elementar analyser vario MICRO cube. | [49,50] |
| Inorganic nitrogen ($N_{In}$), nitrate ($NO_3\_N$) and ammonium ($NH_4\_N$) | Potassium chloride extraction.<br>Analysed using a Skalar SAN++ continuous flow auto-analyser and spectrophotometric detection. | [51–53] |
| Available phosphorus ($P_{av}$) | Olsen's P method.<br>Sodium bicarbonate extraction.<br>Analysed using a Skalar SAN++ continuous flow auto-analyser and spectrophotometric detection | [47,54] |
| Exchangeable potassium (K+) | Ammonium chloride extraction.<br>Analysed using ICP-OES Thermo Fisher ICAP 7600. | [43,55,56] |
| Worm Count (field measure) | Three 15 $cm^3$ holes were dug at a random positions within each 1/3 of each plot. For each, the material was broken apart by hand and all worms were enumerated. | |
| Microbial respiration | Fresh soil incubated with sodium hydroxide for seven days, followed by titration with hydrochloric acid. | [47] |
| Fungal biomass | Measuring the size of fungi observed by direct microscopy of a known dilution of soil.<br><br>25 fields of view are enumerated per sample droplet which is used to estimate the total fungal biomass within the droplet. | Method developed and taught on the microscopy course offered by the Soil Ecology Lab, Hampshire, UK. |

## Plant sampling and laboratory work

Establishment was measured in May 2024, during the final year of the experiment, using a 0.25m$^2$ quadrat. Three locations were sampled per plot; the plot was divided into three equal sections along its length, and a sample was taken from a random location within each of these sections. The outermost rows were avoided to minimise edge effects. The number of plants within the quadrat was counted.

One week before harvest, in August 2024, eight plants were harvested by hand from each plot in order to measure growth metrics. No plants were taken from the endmost one metre, nor from the outermost rows to avoid edge effects, but otherwise the position in the stand was randomised. These samples were used to determine plant height, tiller count and number of grains-per-head, before they were oven dried at 40°C to constant weight to measure dry biomass.

A visual assessment of disease and damage to the grain heads was also performed, and each plot batch was qualitatively classified as having none (0% of heads affected), mild (<20% affected), moderate (20–40% affected) or severe (>40% affected) disease/damage losses (S1 Fig). This is because head losses to damage and disease are a contributing factor to lower overall yields.

Total yield measures for the plots were determined at harvest in August 2024, when grain moisture content was 17%. A sample of each yield batch was then dried for 24 hours to a moisture content of 12–14%, which was used for the thousand grain measurement.

### Statistical analysis

Data were recorded and cleaned in Microsoft Excel, which was also used to determine median and range values. Graphs were created in R Studio version 4.3.2.

Due to the small sample size of just three treatment replicates, the median was used as the most appropriate measure of central tendency, and only descriptive statistics are reported. Differences and trends are described and offer interesting avenues for further research.

For each plot, multiple raw values were measured to determine the plot median ($M_{plot}$). These three plot medians were then used to inform the treatment median ($M_{treatment}$). In tables, these treatment medians are reported as $M_{treatment}$ [Lowest $M_{plot}$, Highest $M_{plot}$] allowing all three $M_{plot}$ values to be easily shown.

The scatter plots show all of the raw data points which inform $M_{plot}$, disaggregated by fertiliser treatment and management type. This shows the entire spread of the dataset, the raw sample size for each parameter, and allows for easy visualisation of any trends, or any location effects of the three trial blocks.

## Results

All raw result data is available in the Supporting Information (S1 Data).

### Year 3 crop results

**Yield.** The spring barley was harvested and weighed at 17% moisture content.

Under conventional management, synthetic fertiliser produced the highest yields (yield $M_{treatment}$ = 1.39t/ha [1.00, 2.49]), followed by humanure (yield $M_{treatment}$ = 1.01t/ha [0.95, 1.47]) then the control (yield $M_{treatment}$ = 0.67t/ha [0.58, 0.88]), with some overlap (Fig 2). Under regenerative management, synthetic fertiliser produced the highest yields (yield $M_{treatment}$ = 4.15t/ha [1.42, 4.94]), followed by humanure (yield $M_{treatment}$ = 2.94t/ha [2.20, 3.10]) then the control (yield $M_{treatment}$ = 0.82t/ha [0.64, 1.24]), however the graph reveals the large variance for SF, and subsequent overlap.

### Other crop growth measures

Other measures of crop growth provide greater insight into the differences observed in overall yields (Table 4). Graphical representation of these results are included in the Supporting Information (S2-S7 Figs).

Under conventional management, the higher SF yield is likely attributable to higher tillering, higher number of grains per head and lower damage losses than humanure and the control. Humanure had the best establishment (Establishment $M_{treatment}$ = 67 plants per 0.25m$^2$ [45,57]) but lower grain number, which lead to a lower head biomass. The control plot performed the lowest across all growth parameters, which correlated with the low overall yield.

Under regenerative management, the higher SF yield is likely attributable to the high number of grains per head (Grain Number $M_{treatment}$ = 24.5 grains per head [16,25]).

The fact that yields were higher under regenerative management than conventional management for all three fertilisation treatments is interesting, considering that the regenerative plots saw lower establishment and lower 1000 grain mass (except for SF treatment), and similar tillering. Grain number was consistently higher in the regenerative plots, and thus appears to be the strongest predictor of yield. This theory is strengthened by the fact that within each management group, the grain number was well correlated with the yield performance for each of the three fertilisers. Additionally, disease and damage was consistently lower under the regenerative management.

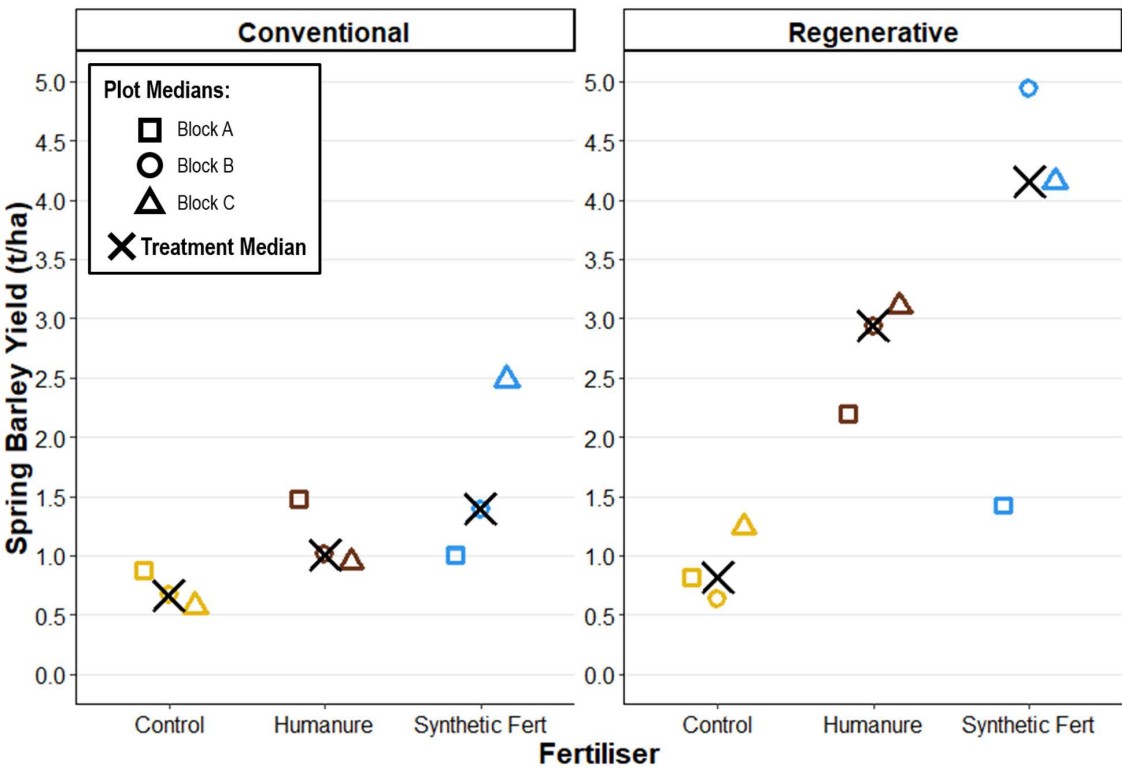

**Fig 2. Spring barley yield at harvest of the third cropping season, under conventional and regenerative management; 17% moisture content.**

**Table 4. Treatment medians for six crop growth parameters for the spring barley harvested in August 2024. Square brackets indicate the lower and higher plot medians. N=the number of raw data points which informed each of these plot medians.**

| Parameter | Unit | Conventional | | | Regenerative | | |
|---|---|---|---|---|---|---|---|
| | | Control | Humanure | SF | Control | Humanure | SF |
| Establishment (n=3) | Plants per 0.25m² | 57 [37,58] | 67 [45, 70] | 57 [44 58] | 49 [40, 53] | 48 [42, 50] | 37 [21, 41] |
| Tillers (n=8 plants) | Tillers per plant | 3 [2.5, 4] | 5.5 [5, 6] | 6 [5, 7] | 6 [4, 6] | 5 [3, 5.5] | 4.5 [4, 7] |
| Height (n=8 plants) | cm | 24.5 [22, 25] | 35.5 [25.5, 41.5] | 42 [37.5, 51.5] | 35 [29.5, 46.5] | 45.5 [38, 55.5] | 62 [43.5, 67] |
| Grain Number (n=8 plants) | Grains per head | 6 [5, 7.5] | 10 [7.5, 15.5] | 16 [10.5, 19.5] | 9.5 [9, 11] | 15 [11, 24] | 24.5 [16, 25] |
| 1000 grain mass (n=1) | g | 44.0 [43.4, 46.8] | 47.0 [43.4, 50.1] | 46.8 [36.1, 50.8] | 38.1 [33.7, 51.7] | 39.3 [37.1, 43.4] | 47.8 [40.3, 51.9] |
| Dry Head Biomass (n=1) | g per head | 0.19 [0.13, 0.26] | 0.21 [0.18, 0.32] | 0.56 [0.43, 1.19] | 0.35 [0.30, 0.45] | 0.63 [0.40, 1.41] | 1.46 [0.50, 1.65] |
| Head damage/ disease (n=1) | Qualitative assessment* | mild+ | moderate+ | mild | mild+ | none+ | none+ |

*The modal disease classification of the three plots is reported. Where one plot was worse than the other two, a '+' is added to the classification.

## Soil

Fifteen parameters of soil health were assessed after three years of cropping. Additional graphical representations of those parameters which are not included in the manuscript are available in the Supporting Information (S8-S10 Figs).

### Physical

**Bulk density.** Bulk density did not change notably throughout the experiment duration, with all treatments showing overlap with the baseline value range in both Year 2 and Year 3 (Table 5; and S8 Fig).

### Water stable aggregates

At the end of Year 3, under conventional management, humanure exhibited a slightly lower median WSA than the control and SF treatments (humanure WSA $M_{treatment}$ = 56.11% [53.54, 58.05]), but there was a high degree of overlap between the different treatment plots, so this difference is not considered meaningful (Fig 3). All nine conventional $M_{plot}$ values fell within 51.9% and 67.41%.

Under regenerative management, wet aggregate stability was again similar between the three fertiliser treatments, with a high degree of overlap between the different treatment plots. Synthetic fertiliser reported the highest median value in plot R04_SF (Block A), but also the greatest range (SF WSA $M_{treatment}$ = 82.33% [69.27, 82.36]). All nine regenerative $M_{plot}$ values fell within 69.27% and 83.33%.

This indicates that fertiliser treatment did not have a meaningful impact on WSA, with a high degree of overlap and similar variance between each fertiliser treatment within each management group. However, there is a clear impact of management on water stable aggregates, with the regenerative practices resulting in higher WSA across all three fertiliser treatments.

There were no clear directional changes in WSA over time, when comparing between Year 2 and Year 3, with some plots showing slight increases and others decreases (Table 6). Most notably, regenerative management displayed considerably higher WSA than conventional management in both years.

### Chemical

**pH.** There were no clearly observed trends over time between the baseline, Year 2 and Year 3 of data collection, indicating a broadly stable soil pH (Table 7).

In Year 3, all $M_{plot}$ values fell within the recorded range of the baseline soil (6.98–7.55), except for plot R08_Control (Block B) where $M_{plot}$ = 7.56 (Fig 4). There was no clearly observed trend between treatments within each management

**Table 5. Treatment medians for topsoil bulk density (g/cm³ oven dry soil), measured at a depth of 5-10 cm for the baseline soil, and for Year 2 and Year 3. Square brackets indicate the lower and higher plot medians. N = the number of raw data points which informed each of the plot medians.**

|  | Conventional | | | Regenerative | | |
|---|---|---|---|---|---|---|
|  | Control | Humanure | SF | Control | Humanure | SF |
| Baseline‡ (Year 0) (n = 3) | 1.54 [1.48, 1.56] | | | | | |
| Year 2* (n = 2) | 1.49 [1.41, 1.55] | 1.44 [1.41, 1.70] | 1.45 [1.38, 1.49] | 1.41 [1.41, 1.50] | 1.49 [1.49, 1.55] | 1.63 [1.52, 1.64] |
| Year 3* (n = 2) | 1.51 [1.47, 1.52] | 1.52 [1.50, 1.55] | 1.54 [1.47, 1.55] | 1.53 [1.42, 1.53] | 1.51 [1.44, 1.58] | 1.48 [1.45, 1.55] |

‡ n = 3; Reported as median [lowest value, highest value].

* n = 2; 2 samples taken from each plot, combined to give the median result for each plot. The three plot results are combined to give the overall treatment median, and are reported as median plot result [lower plot result, higher plot result].

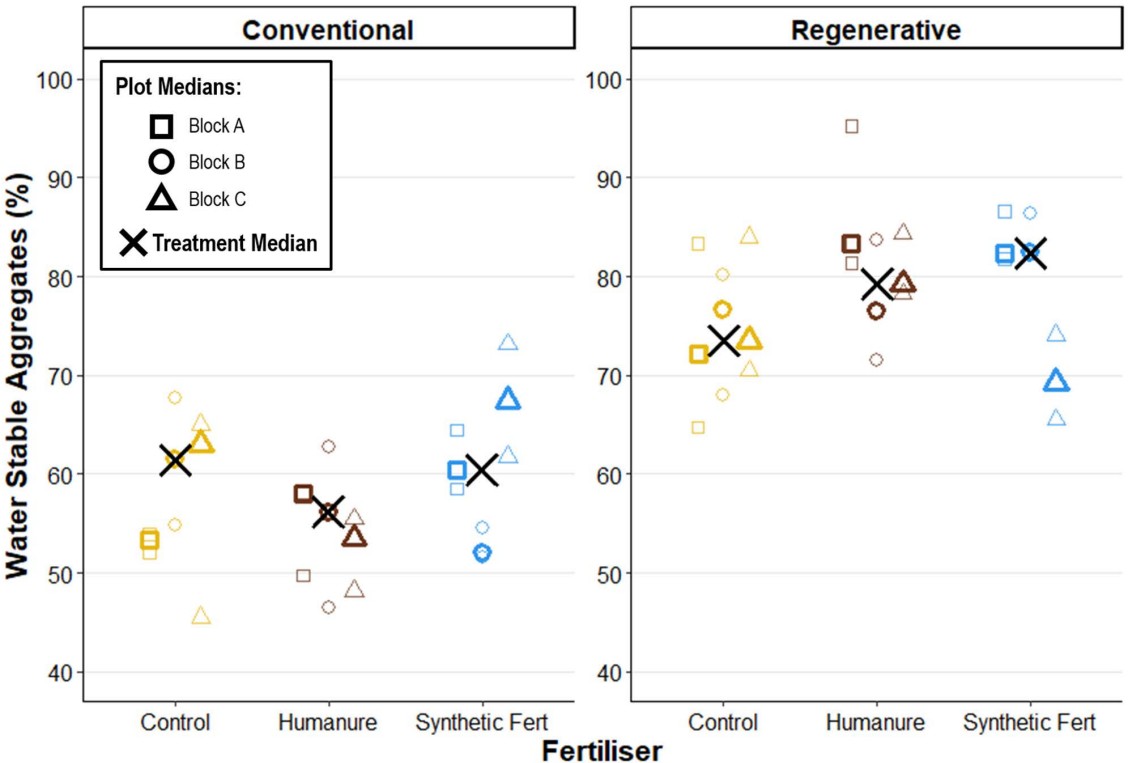

**Fig 3. Water stable aggregates after three years of cropping.** Percentage of air-dried particles retained on a 1 mm sieve after submersion and agitation. Raw data points are displayed to show the degree of variation informing each plot median.

type, nor between the management types. Within-plot replication data clustering was high, increasing confidence in the results. Within-treatment clustering was also high, except for plot R03_Control (Block A) which was much lower than the rest of Block A and the rest of the Regenerative-Control plots.

## Moisture

Moisture content reflects a snapshot of the field conditions on one given day, and is primarily used in lab calculations to convert between wet and dry masses. As such, it was not used to make comparisons over time, but instead to make

**Table 6. Plot medians for water stable aggregates (%) in topsoil after two and three years of cropping, sampled at a depth of 0-15 cm. The treatment median is shaded for easy reference.**

|  | Conventional | | | | | | | | | Regenerative | | | | | | | | |
|---|---|---|---|---|---|---|---|---|---|---|---|---|---|---|---|---|---|---|
|  | Control | | | Humanure | | | SF | | | Control | | | Humanure | | | SF | | |
| Block | A | B | C | A | B | C | A | B | C | A | B | C | A | B | C | A | B | C |
| Year 2* | 59 | 74 | 56 | 57 | 60 | 61 | 62 | 53 | 54 | 64 | 89 | 86 | 67 | 77 | 69 | 63 | 96 | 79 |
| Year 3* | 53 | 61 | 63 | 58 | 56 | 54 | 60 | 52 | 67 | 72 | 77 | 73 | 83 | 76 | 79 | 82 | 82 | 69 |

*n = 3; 5 grab samples taken from each plot, compiled into one mixed bulk sample, then lab test conducted in triplicate. The median of these lab tests gave the result for each plot.*

**Table 7. Treatment medians pH in topsoil at the baseline, and after two and three years of cropping, sampled at a depth of 0-15 cm. Square brackets indicate the lower and higher plot medians. N = the number of raw data points which informed each of the plot medians.**

| | Conventional | | | Regenerative | | |
|---|---|---|---|---|---|---|
| | Control | Humanure | SF | Control | Humanure | SF |
| Baseline‡ (Year 0) | 7.49 [6.89, 7.55] | | | | | |
| Year 2* (n = 3) | 7.30 [6.84, 7.31] | 7.63 [7.50, 7.71] | 7.39 [7.32, 7.42] | 7.23 [6.83, 7.32] | 6.82 [6.70, 6.99] | 7.19 [7.08, 7.36] |
| Year 3 (n = 3) | 7.05 [6.96, 7.15] | 7.29 [7.27, 7.44] | 7.16 [7.15, 7.21] | 7.54 [6.90, 7.56] | 7.27 [7.25, 7.27] | 7.39 [7.31, 7.48] |

‡ n = 3; 5 grab samples taken from each plot, compiled into one mixed bulk sample, then lab test conducted in triplicate. The median of these lab tests gave the result for each plot.

*n = 3; 5 grab samples taken from each plot, compiled into one mixed bulk sample, then lab test conducted in triplicate. The median of these lab tests gave the result for each plot. The three plot results are combined to give the overall treatment median, and are reported as median plot result [lower plot result, higher plot result].

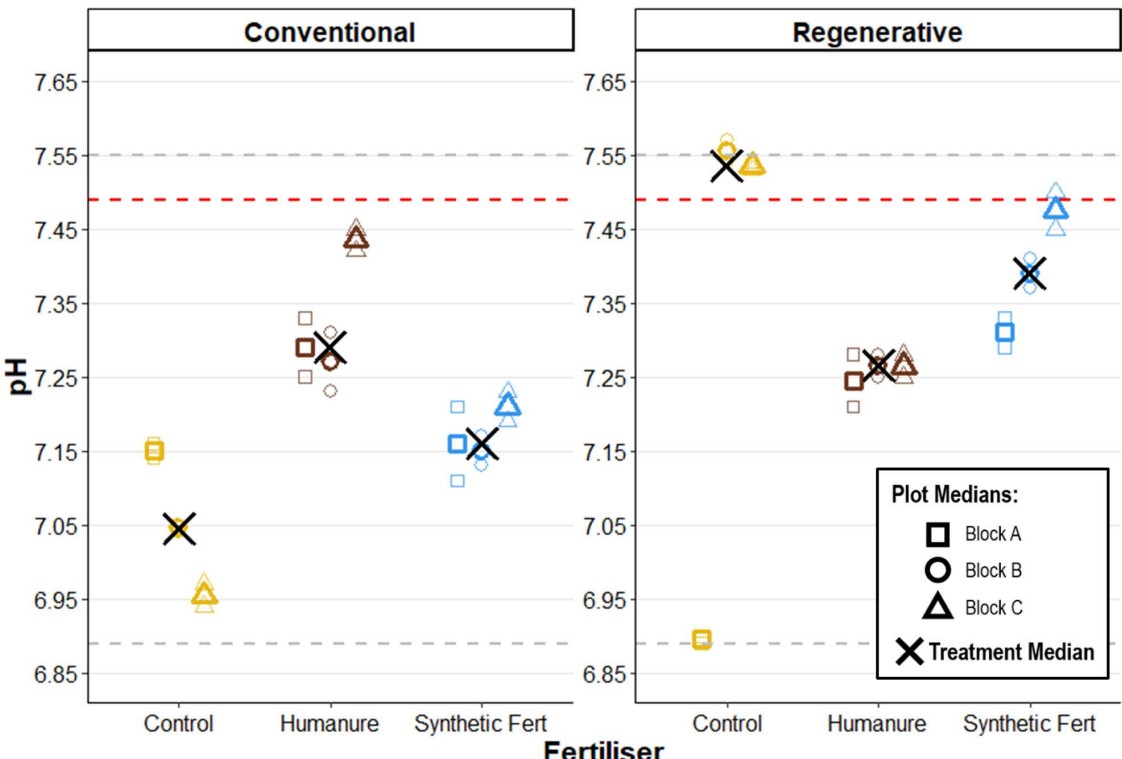

**Fig 4. Soil pH after three years of cropping.** The red line shows the baseline soil median value, and the grey lines show the baseline soil value range. Raw data points are displayed to show the degree of variation informing each plot median.

internal comparisons at the end of Year 3. No clear location-based effects are observed across the field, from Block A (northwest) to Block C (southeast) (Fig 5).

Under conventional management, there are no clear differences between the three fertiliser treatments. Under regenerative management a clearer difference can be observed due to smaller variance, and with humanure plots

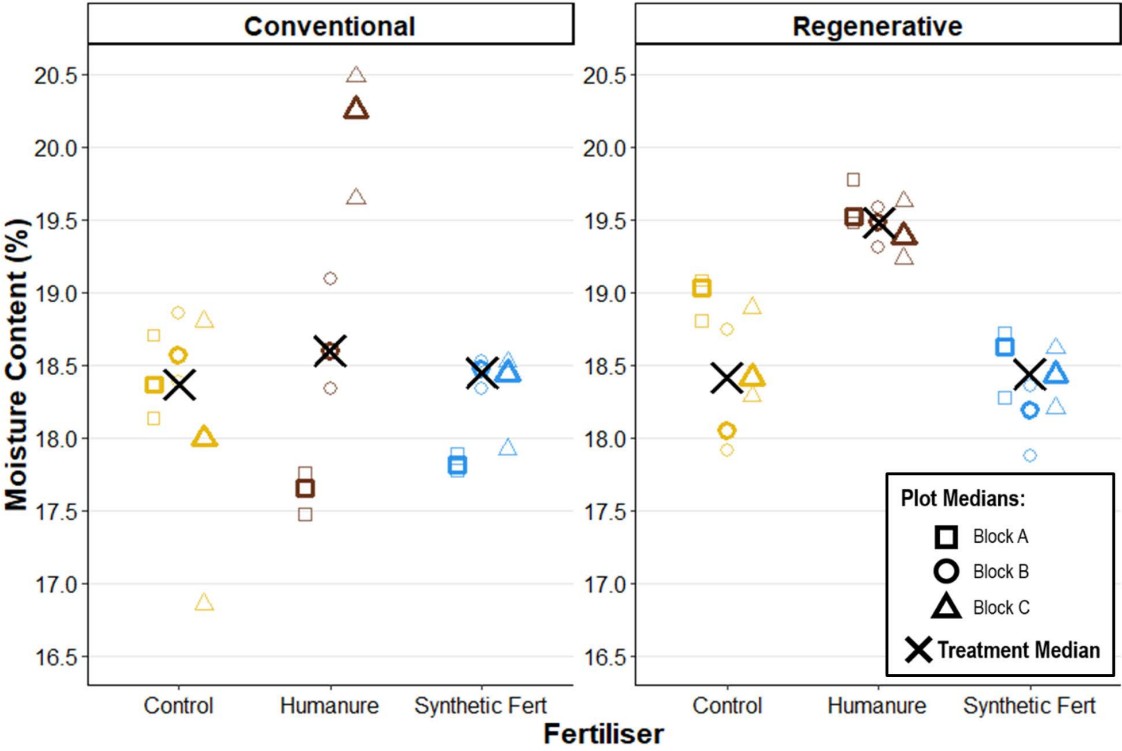

**Fig 5. Soil moisture content measured at the end of Year 3.** Raw data points are displayed to show the degree of variation informing each plot median.

displaying a higher median moisture content than SF and control treatments (humanure Moisture $M_{treatment}$ = 19.48% [19.38, 19.52]).

## No clear difference is observed between management types

**Organic matter and Carbon.** Regenerative plots saw a slight increase in SOM between the baseline and Year 2, though this was less pronounced in Year 3 (Table 8). The conventionally managed humanure plot followed this same pattern, with the other conventionally managed treatments showing a slight decrease.

After three years of cropping under conventional management, humanure resulted in the highest SOM (humanure SOM $M_{treatment}$ = 4.87% [4.75, 5.54]), with the median exceeding the upper range of the baseline soil (baseline SOM $M_{treatment}$ = 4.70% [4.63, 4.85]) (Fig 6). Both the control and SF treatments showed very similar SOM values, which were below the baseline SOM range, indicating a small decline.

Under regenerative management, humanure clearly showed a higher SOM value than the other two fertiliser treatments (humanure SOM $M_{treatment}$ = 5.28% [5.08, 5.33]), and well exceeded the baseline SOM value range, indicating a meaningful increase. The control performed slightly better than the SF treatment, and both were within the upper range of the baseline soil.

## Nitrogen

All treatments bar conventional-control raised the Total Elemental Nitrogen ($N_T$) in soil after three years, compared to the baseline value (Fig 7). High inter-plot variability within each treatment meant that no clear effect of management nor fertiliser type could be observed, suggesting that they are not significant predictors of $N_T$.

**Table 8. Treatment median soil organic matter (%) in topsoil at the baseline, and after two and three years of cropping, sampled at a depth of 0-15 cm. Square brackets indicate the lower and higher plot medians. N = the number of raw data points which informed each of the plot medians.**

|  | Conventional | | | Regenerative | | |
|---|---|---|---|---|---|---|
|  | Control | Humanure | SF | Control | Humanure | SF |
| Baseline‡ (Year 0) | 4.70 [4.63, 4.85] | | | | | |
| Year 2* (n = 3) | 4.79 [4.34, 4.85] | 5.17 [4.69, 5.26] | 4.72 [4.55, 4.82] | 5.26 [4.92, 5.40] | 5.33 [4.93, 5.64] | 5.19 [5.17, 5.22] |
| Year 3* (n = 3) | 4.56 [4.44, 4.61] | 4.87 [4.75, 5.54] | 4.54 [4.38, 4.63] | 4.82 [4.74, 5.01] | 5.28 [5.08, 5.33] | 4.73 [4.68, 4.97] |

‡ *n = 3; 5 grab samples taken from each plot, compiled into one mixed bulk sample, then lab test conducted in triplicate. The median of these lab tests gave the result for each plot.*

\* *n = 3; 5 grab samples taken from each plot, compiled into one mixed bulk sample, then lab test conducted in triplicate. The median of these lab tests gave the result for each plot. The three plot results are combined to give the overall treatment median, and are reported as median plot result [lower plot result, higher plot result].*

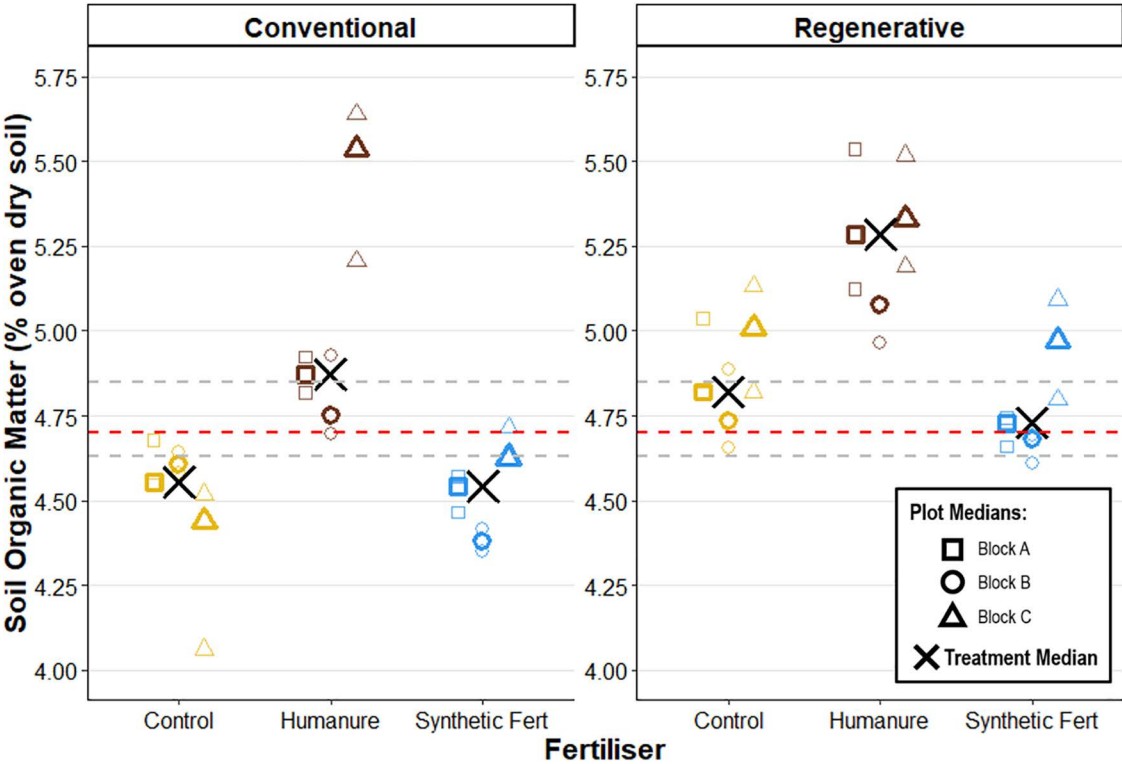

**Fig 6. Soil organic matter content after three years of cropping.** The red line shows the baseline soil median value, and the grey lines show the baseline soil value range. Raw data points are displayed to show the degree of variation informing each plot median.

Inorganic forms of N are highly transient within the soil system, and so comparisons over time are not meaningful. Instead, comparisons between the plots can show differences in snapshot values in a given moment. Inorganic N ($N_{In}$) only comprised around 0.1% of the total elemental N ($N_T$) in the soils, and $M_{treatment}$ values ranged from 2.10–5.44 mg/kg (Fig 8).

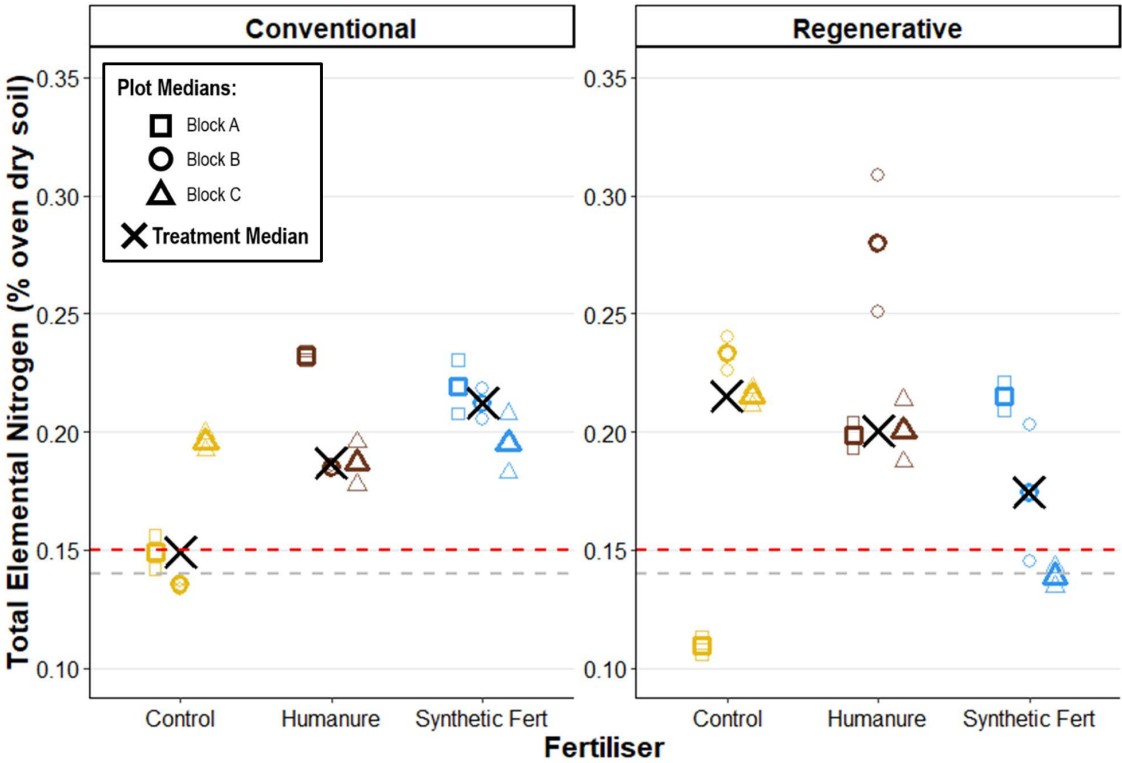

**Fig 7. Soil total elemental nitrogen content after three years of cropping.** The red line shows the baseline soil median value, and the grey lines show the baseline soil value range. Raw data points are displayed to show the degree of variation informing each plot median.

Similarly to $N_T$, after three years under conventional management $N_{In}$ was elevated in the humanure and SF fertiliser treatments as compared with the control treatment. This trend was not observed under regenerative management, where all three treatments showed high intra and inter plot variation, with a high degree of overlap. The lack of a clear directional effect of management type or fertiliser type on $N_{In}$ suggests that they are not significant predictors of $N_{In}$.

Disaggregating inorganic N into its two major forms, we observe that nitrate N ($NO_3^-\_N$) comprised a large proportion of $N_{In}$ (typically > 75%) (Table 9), and subsequently followed a very similar distribution (S9 Fig). No clear directional effect of management type nor fertiliser type is observed.

Ammonical N ($NH_4^+\_N$) comprised a smaller proportion of $N_{In}$ than nitrate. All treatments across both management types resulted in $NH_4\_N$ concentrations below the baseline soil range (baseline $NH_4^+\_N$ $M_{treatment}$ = 1.42 mg/kg [1.34, 1.57]) (S10 Fig). No meaningful differences were observed between groups, with all $M_{treatment}$ values falling within 0.61–0.83 mg/ kg, and no clear effect of management type nor fertiliser treatment was observed.

### Available phosphorus

After three years under both conventional and regenerative management, humanure showed elevated available phosphorus ($P_{av}$) compared with the other fertilisers (conventional humanure $P_{av}$ $M_{treatment}$ = 17.97 mg/kg [12.95, 25.96]; regenerative humanure $P_{av}$ M = 16.72 mg/kg [8.63, 23.80]) (Fig 9). Both management types also showed a high degree of variation between the humanure plots, whereas the control and SF plot results were better clustered.

Regenerative treatments showed slightly lower $P_{av}$ than their conventional comparators, but this difference does not appear to be meaningful. Humanure fertiliser appears to raise $P_{av}$ to within the Index 2 range, the recommended range for arable crops [58].

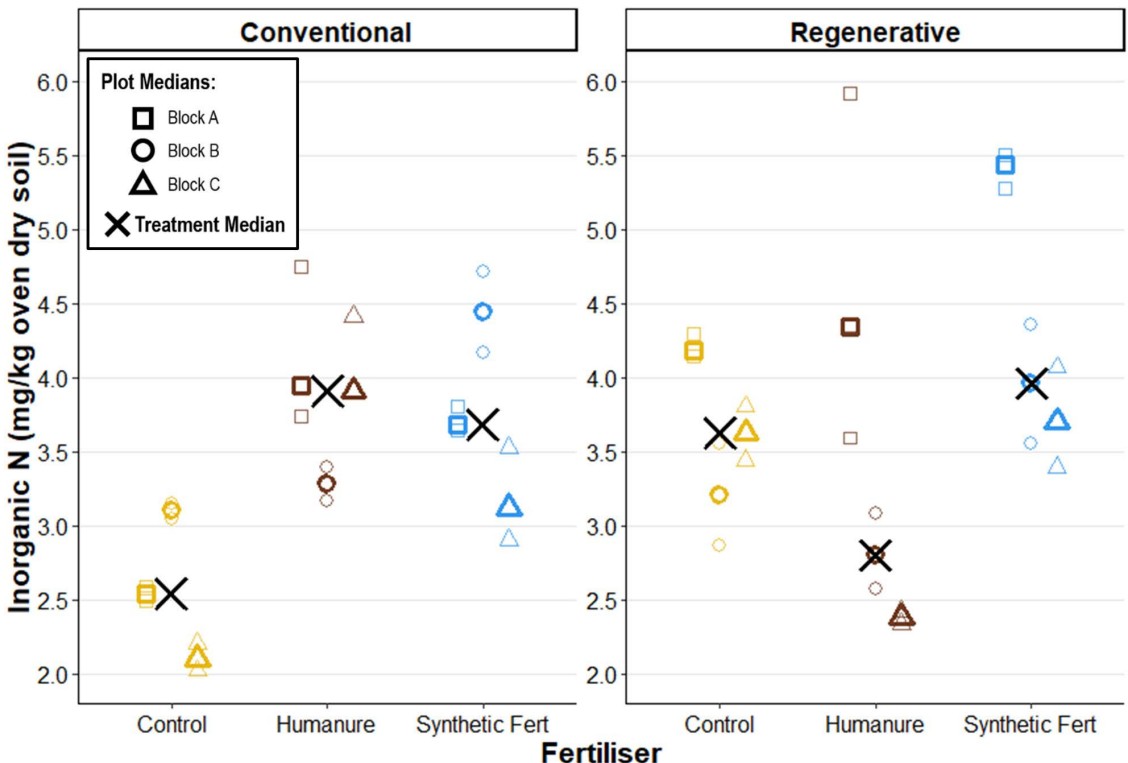

**Fig 8. Soil inorganic nitrogen content after three years of cropping.** Raw data points are displayed to show the degree of variation informing each plot median.

**Table 9. Treatment medians for topsoil inorganic nitrogen (mg/kg oven dry soil), measured at a depth of 5-10 cm after three years of cropping.** Square brackets indicate the lower and higher plot medians. N = the number of raw data points which informed each of the plot medians.

| | Conventional | | | Regenerative | | |
|---|---|---|---|---|---|---|
| | Control | Humanure | SF | Control | Humanure | SF |
| **Total Inorganic N** (n=3)* | 2.54 [2.10, 3.10] | 3.91 [3.28, 3.94] | 3.68 [3.12, 4.44] | 3.62 [3.21, 4.18] | 2.80 [2.38, 4.35] | 3.96 [3.70, 5.44] |
| **NH$_4$_N** (n=3)* | 0.63 [0.55, 0.67] | 0.83 [0.72, 0.88] | 0.61 [0.58, 0.70] | 0.75 [0.69, 0.86] | 0.76 [0.63, 1.52] | 0.77 [0.66, 0.80] |
| **NO$_3$_N** (n=3)* | 1.82 [1.47, 2.46] | 2.97 [2.54, 3.19] | 3.04 [2.44, 3.96] | 2.64 [2.42, 3.42] | 2.08 [1.78, 2.75] | 3.23 [2.83, 4.67] |

*n=3; 5 grab samples taken from each plot, compiled into one mixed bulk sample, then lab test conducted in triplicate. The median of these lab tests gave the result for each plot. The three plot results are combined to give the overall treatment median, and are reported as median plot result [lower plot result, higher plot result].*

## Potassium

After three years under conventional management, humanure elevated potassium cations (K+), as compared with the control and SF treatments (humanure K + M$_{treatment}$ = 149.79 mg/kg [149.67, 201.61]) (Fig 10). This was also true under regenerative management (humanure K + M$_{treatment}$ = 174.65 [109.51, 175.75]), although one humanure plot (R09_Humanure (Block C)) was significantly lower than the other two, overlapping completely with the control and SF treatments and thus making the difference between the treatments less certain.

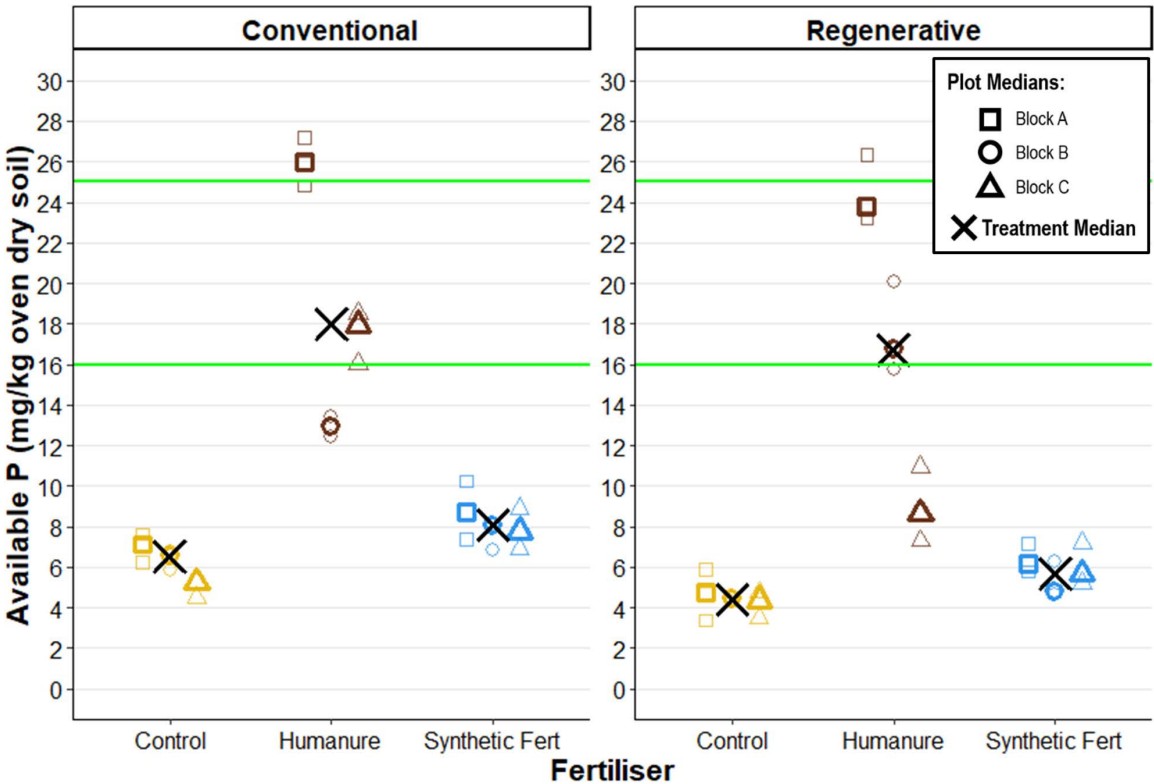

**Fig 9. Soil available phosphorus after three years of cropping.** The green lines show the boundaries for Index 2, the recommended range for arable crops. Raw data points are displayed to show the degree of variation informing each plot median.

$M_{treatment}$ values for humanure fertiliser fell within the Index 2- value range, the recommended range for arable crops [58], with some individual plots falling outside of this range. Under conventional management, the control and SF treatment plots were generally just below this range, but were just within this range under regenerative management.

## Biological

**Worms.** After three years under conventional management, worm count was highest in the humanure treatment plots (humanure Worms $M_{treatment}$ = 13 [12,13]), though the SF and control treatments were not considerably lower (SF Worms $M_{treatment}$ = 11 [9,13]; control Worms $M_{treatment}$ = 11 [9,12]) (Fig 11).

Under regenerative management this difference was even more pronounced, with humanure achieving a median worm count of $M_{treatment}$ = 23 [22,23]. These findings indicate that fertilisation type is a significant predictor of worm count, with a strong positive effect associated with humanure addition.

Worm counts were significantly higher for all treatments under regenerative management than conventional management, indicating that management type is a significant predictor of worms, with a strong positive effect associated with regenerative management.

## Respiration

After three years of cultivation under conventional management, humanure treatment resulted in the highest respiration rate (humanure respiration $M_{treatment}$ = 1.66g $CO_2$ kg dry soil$^{-1}$ y$^{-1}$[1.66, 1.75]), followed by SF treatment (SF respiration

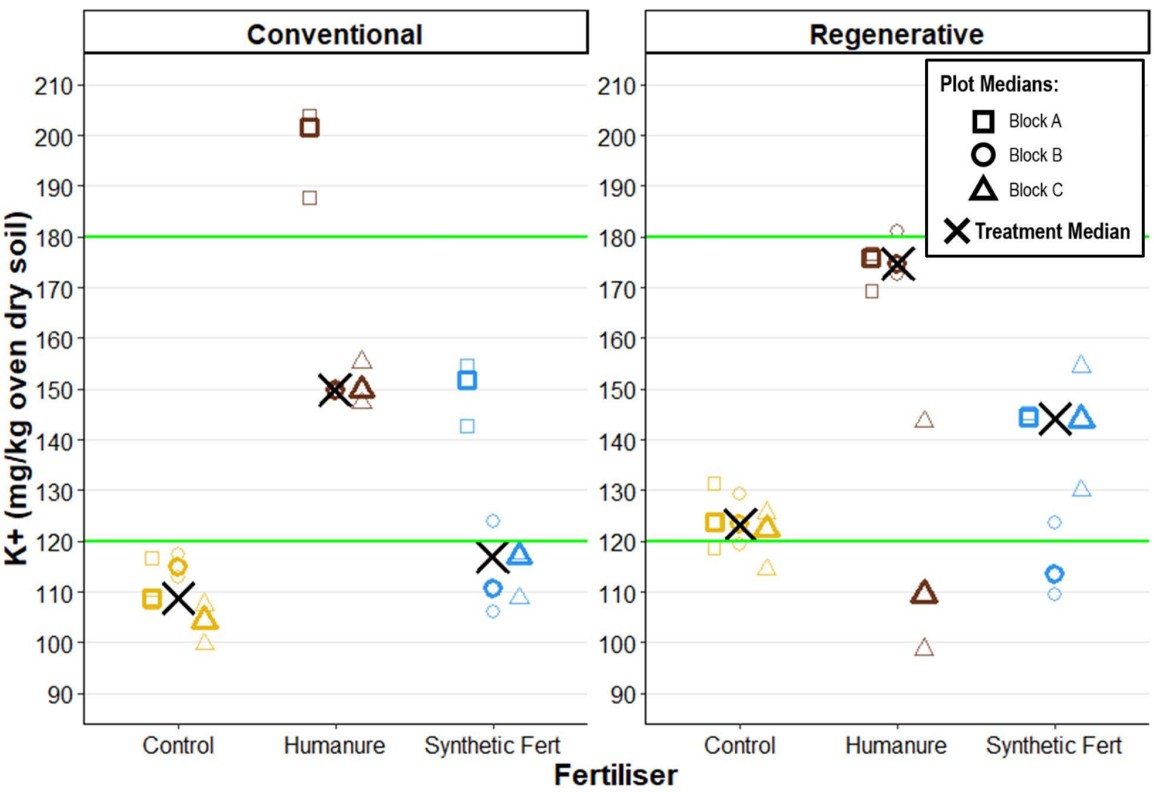

**Fig 10. Soil potassium after three years of cropping.** The green lines show the boundaries for Index 2-, the recommended range for arable crops. Raw data points are displayed to show the degree of variation informing each plot median.

$M_{treatment}$ = 1.23g $CO_2$ kg dry soil$^{-1}$ y$^{-1}$ [1.09, 1.42]) (Fig 12). The low degree of overlap indicates a significant positive effect of humanure on respiration rate under conventional management.

Under regenerative management humanure treatment also resulted in the highest respiration rate (humanure respiration $M_{treatment}$ = 1.73g $CO_2$ kg dry soil$^{-1}$ y$^{-1}$[1.46 1.95]), but the effect is less pronounced due to a high degree of overlap with SF treatment (SF respiration $M_{treatment}$ = 1.57g $CO_2$ kg dry soil$^{-1}$ y$^{-1}$ [1.41, 1.88]).

Regeneratively managed plots saw higher respiration rates than their conventionally managed comparators, though this effect was weaker under humanure fertilisation. These findings suggest a positive effect of humanure and of regenerative management on soil microbial respiration rate, though the magnitude of these effects were variable.

### Fungal biomass

After three years under conventional management, humanure treatment resulted in the highest median values of fungal biomass (humanure Fungi $M_{treatment}$ = 5.87 µg ml$^{-1}$ air dry soil [4.10, 6.21]). The control plots showed extreme variation (control Fungi M = 2.07 µg ml$^{-1}$ air dry soil [0.00, 15.53]) with very high values reported in plot C09_Control (Block C), obscuring any clear positive effect of humanure application (Fig 13).

Under regenerative management a similar trend is observed, with humanure reporting the highest median fungal biomass, and the highest reported raw values (humanure Fungi $M_{treatment}$ = 16.18 µg ml$^{-1}$ air dry soil [12.66, 20.70]). However, there remained a significant degree of overlap between the treatment results.

The graph reveals a positive effect of regenerative management on fungal biomass, as compared with conventional management, as well as a much lower number of null (zero) results observed under regenerative management. Direct

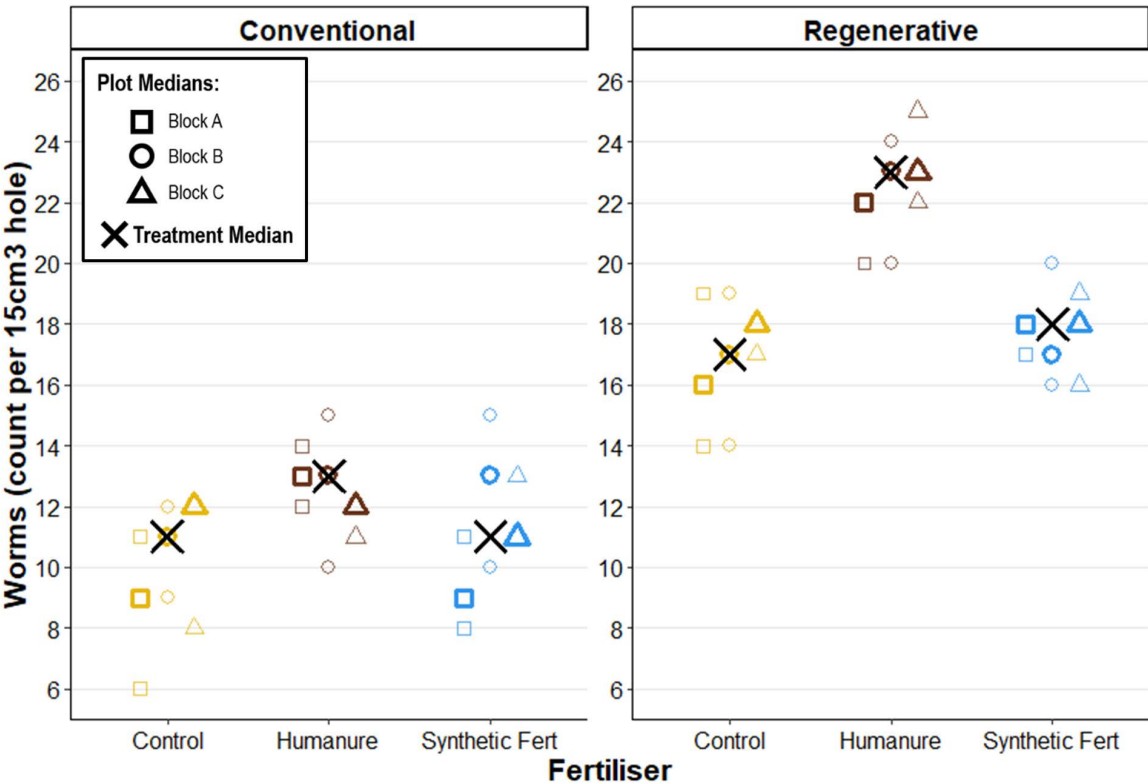

**Fig 11. Total worm count per 15 cm³ hole after three years of cropping.** Raw data points are displayed to show the degree of variation informing each plot median.

comparison of each fertiliser within each Block showed that $M_{plot}$ fungal biomass was higher under regenerative management in all nine cases.

These findings indicate that management type is a significant predictor of fungal biomass, with a clear positive effect of regenerative management. The effect of fertiliser type is less clear, although humanure appears to have a promising positive effect, particularly when compared with synthetic fertiliser.

## Discussion

### Crop growth parameters

**Humanure in conventional management.** Under conventional management, humanure treatment resulted in a small increase in crop yield as compared to the unfertilised control, with an improvement of 0.34t/ha between the median values. This suggests a weakly positive fertilisation effect and is consistent with other studies which found that humanure increased crop yields compared to no fertilisation [15–17]. However, the yield from the synthetic fertiliser (SF) treatment was higher still, improving upon the humanure treatment by an additional 0.38 t/ha. These results indicate that the direct substitution of recommended synthetic fertiliser rates with humanure does not result in equivalent crop yields within a three-year timeframe.

These results should be interpreted with caution due to small sample size and the relatively large variation observed in the SF treatment. Moreover, all reported yields were markedly lower than the 2024 Yorkshire regional average of 5.2 t/ha

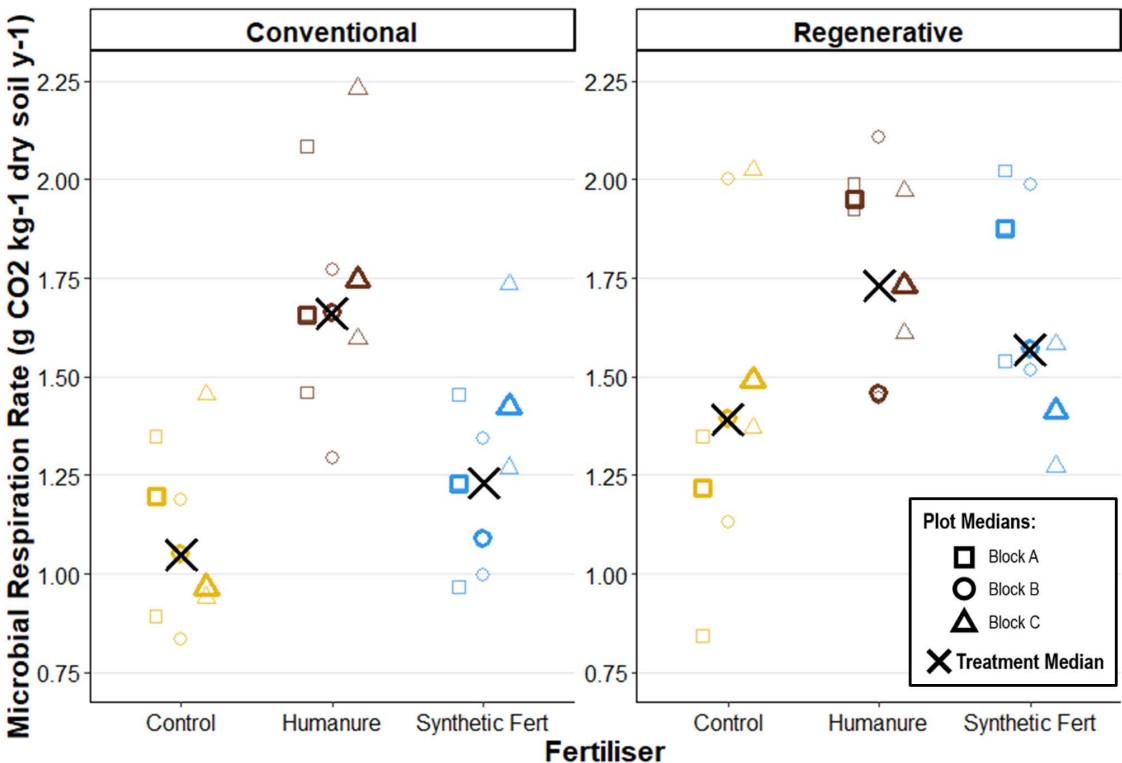

**Fig 12. Soil microbial respiration rate after three years of cropping.** Raw data points are displayed to show the degree of variation informing each plot median.

[59], including the synthetic fertiliser treatment, which is typically considered the "best practice" benchmark. This general underperformance raises the possibility that other environmental or management factors may have constrained yield across all treatments. One possibility is bird damage, which was noted in June. Additionally, the markedly wet and low-sun conditions of April and May compared to the previous five years [60], which are critical months for ear development, as well as wet weather in September which delayed harvest, could have led to yield losses from disease and rot. Finally, the land transition from grassland to arable land may initially constrain yields, as compared to established arable systems [61].

Analysis of individual crop performance parameters offers further insight into these yield patterns. The humanure plots exhibited better plant establishment, greater tillering, higher grain number and higher grain mass compared to the control, which contributed to the yield increase over the control. Interestingly, while SF plots showed poorer establishment than the humanure plots, they had a markedly higher number of grains per head, which appears to be the primary factor driving their higher overall yield. This indicates that grain number was the most significant determinant of yield under conventional management. Additionally, grain loss due to damage and disease was higher in the humanure plots than the control and SF plots, and could have led to a slightly lower yield than would have been expected from the establishment and growth measures.

These findings are consistent with the AHDB Barley Growth Guide [44] which suggests that final yield is more strongly influenced by grain number than grain size. The guide also highlights a potential negative correlation between the number of grains per plant and plant density (plants/m²), a pattern that is also observed in the present study.

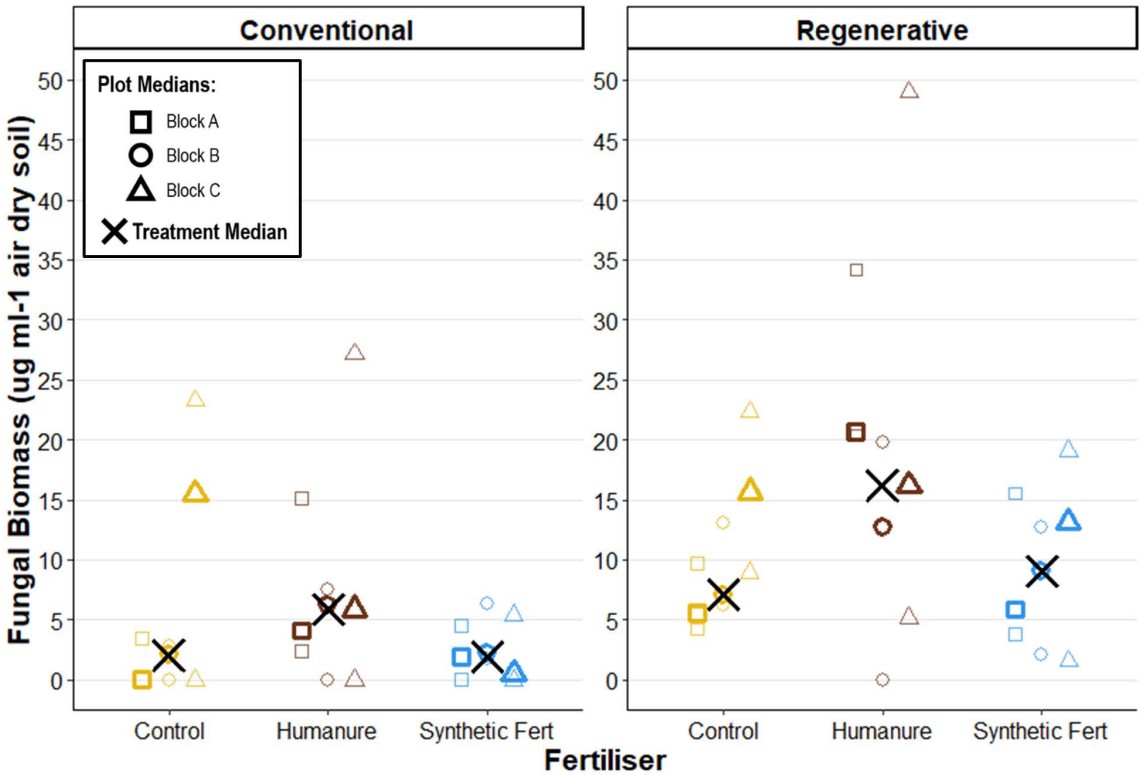

**Fig 13. Soil fungal biomass after three years of cropping, calculated from direct enumeration and measurement via light microscopy.** Raw data points are displayed to show the degree of variation informing each plot median.

Barley, particularly two-row varieties such as in the present study, is characteristically sink-limited. This means that it often has insufficient grains to store all assimilate formed during grain filling [62]. As such, early developmental processes that influence ear and grain number are especially important for yield.

Synthetic fertilisers may therefore be better equipped to support these critical early growth stages by supplying nitrogen in readily available forms at precisely the right time. This capacity for targeted nitrogen delivery offers a distinct advantage over the slower, less predictable release from organic amendments like humanure. This finding underscores the role of optimised nutrient timing and formulation in supporting high-yielding, intensively bred cereal crops in a conventionally managed system.

### Humanure in regenerative management

Under regenerative management a similar trend was observed, with humanure fertilisation improving yield by 2.12t/ha compared to the unfertilised control. This substantial increase provides clear evidence that humanure can deliver meaningful fertilisation benefits to spring barley. The fact that these benefits emerged within just three years is encouraging and suggests that humanure can become agronomically effective within a relatively short time frame when managed appropriately.

Although the median yield for the synthetic fertiliser treatment was higher than that of humanure (SF $M_{treatment}$ = 4.15 t/ha [1.42, 4.94]; humanure $M_{treatment}$ = 2.94 t/ha [2.20, 3.10]), there was considerable overlap in yield values between treatments. Notably, all three humanure plots outperformed plot R04_SF in Block A, which had particularly poor performance

and contributed to the high variance in the SF treatment. This overlap reduces the likelihood of a statistically significant difference between the two treatments and suggests that, under regenerative management, humanure may begin to approach the performance of synthetic fertiliser. These findings indicate the potential for humanure to serve as a viable alternative nutrient source in regenerative cropping systems without compromising yield.

Analysis of individual crop performance parameters revealed that humanure-treated plots exhibited similar establishment, tillering, and grain mass to the control. However, they showed a markedly higher number of grains per head, indicating that grain number was the key driver of yield improvement, which aligns with the findings from the conventional plots.

The synthetic fertiliser treatment once again showed lower establishment than humanure but achieved both a higher grain number per head and a greater grain mass. The lowest median grain number and grain mass values within the SF treatment were recorded in plot R04_SF — the same plot with the lowest overall yield — further substantiating the strong relationship between grain number and yield performance.

## Comparing conventional and regenerative management

Across all fertiliser treatments, the regenerative plots consistently produced higher yields than their conventional counterparts in terms of yield. This is a particularly noteworthy result, as it challenges the prevailing narrative that conventional management delivers the highest yields, while regenerative approaches, though environmentally beneficial, typically do so at the expense of productivity.

Most strikingly, both the humanure and SF treatments achieved higher median yields under regenerative management than the SF treatment under conventional management, which is generally regarded as a "best practice" benchmark. This result provides strong evidence that regenerative farming practices can enhance yield potential, not just environmental outcomes. It represents a compelling case for the wider adoption of regenerative agriculture in temperate cereal cropping systems.

While crop establishment was generally lower under regenerative management, this did not appear to constrain final yield. Tillering was comparable between systems, and although grain mass was lower for the control and humanure treatments under regenerative conditions, it was similar between systems for the synthetic fertiliser treatment. Overall, grain number was higher under regenerative management, reinforcing its positive impact on yield.

Interestingly, a higher incidence of diseased or damaged heads was observed in the conventional plots, which may have contributed to the lower yields. This finding is counterintuitive, given that the conventional system received fungicide applications, whereas the regenerative plots had never been treated. Although this observation could not be analysed in statistical detail, it lends anecdotal support to the idea that regenerative systems, with their greater ecological diversity, may offer a degree of disease suppression through enhanced biological resilience.

Whilst literature is inconclusive about whether organic amendments improve yields versus synthetic fertilisers [63–66] it is likely the case that the potential advantages are highly context specific. The findings of the present study support the hypothesis that humanure is better suited to regenerative systems rather than serving as a direct substitute for synthetic fertilisers in conventional systems. This aligns with the theory that regenerative practices improve microbial functions and associated nutrient cycling [67], including the mineralisation of nutrients in organic residues as they decompose, making them plant-available [68]. It underscores the importance of soil health in driving plant productivity and challenging the conventional wisdom that intensive practices and chemical inputs are required for yield maximisation.

## Soil health

**Soil organic matter and Carbon.** Soil organic matter (SOM) and total carbon ($C_{tot}$) are related measures which are highly correlated [69]. Grassland typically contains higher SOM than arable land [70] due to disturbance-associated losses [57,20], and so a decline from baseline levels was anticipated.

After three years of application, humanure fertilisation resulted in consistently higher SOM than both the control and the SF treatment across both management systems, and raised the SOM content above that which was recorded in the baseline soil assessment at the start of the experiment. Additionally, regenerative management resulted in higher SOM than conventional management across all three fertilisation treatments. These findings indicate that both humanure application and regenerative management have positive effects on SOM, even when transitioning from grassland to arable land. These findings are attributed to the direct addition of organic material to the soil, and the extensive evidence for the benefits of low tillage [71,72] and maintaining living roots in the soil [73,74] exemplified in the regenerative plots.

The control and SF treatments did not ever exceed the range recorded in the baseline soil assessment, indicating that they had no positive effect on SOM, which is expected due to no direct organic matter addition. Under conventional management both the control and SF treatments actually saw a slight reduction in SOM compared to the baseline, which supports the theory that practices like heavy tillage accelerate the breakdown of SOM and release it as $CO_2$ [72,75].

Soil organic matter offers several well-established benefits to soil health. It improves soil structure, enhances water holding capacity, helps regulate temperature, and provides both food for soil microbes and a slow-release source of nutrition for plants [35,76–78]. These benefits collectively contribute to the long-term health and productivity of the soil ecosystem.

While the increase in SOM observed in this study is a positive indicator of broader ecological and agronomic benefits, the retention of this organic matter in the soil system is not straightforward. Soil carbon dynamics are highly complex, and pose challenges for determining changes in long-term storage versus shorter-term fluxes [79,80]. A more detailed analysis of long-term organic matter retention and the specific carbon fractions involved would be necessary to fully understand the persistence and behaviour of the added material, which was beyond the scope of this work.

## Water stable aggregates

Aggregate stability refers to a soil's resistance to structural breakdown, and is an important indicator of soil health. Low aggregate stability may indicate a soil that is more prone to erosion. Slaking, a specific type of breakdown, occurs when larger air-dry soil aggregates are suddenly immersed in water, and the aggregates are not strong enough to withstand the internal stresses induced by rapid water uptake. This experiment looked at sudden immersion and agitation aggregates in water. Grassland typically contains higher aggregate stability than cultivated land [70,81], with tillage a key driver of aggregate destruction [82].

The difference in water-stable aggregates (WSA) between conventional management and regenerative management was one of the most striking and significant findings of this study, with regenerative plots containing 12–23% higher WSA across all three fertiliser treatments.

WSA is influenced by organic matter, which plays a critical role in improving aggregate stability by binding mineral particles into larger, more stable aggregates and by slowing the rate of wetting [83]. Soil ecology can also contribute to aggregate formation, with 'sticky' root exudates 'gluing' soil particles together [84] and promoting fungal growth which also bind particles together [85]. SOM, fungal biomass and microbial respiration were elevated under regenerative management, which is concordant with these expectations. This also aligns with understanding that over time repeated tillage can break down these aggregates [82,86].

Interestingly, WSA saw no clear effect of fertiliser type, despite the fact that humanure treatment is the direct addition of organic material to the soil. It is anticipated that this is due to the longer time frame required for the surface-applied SOM to be incorporated into the soil, and the slow formation of these aggregates [87,88].

## Nitrogen

Whilst soil nitrogen is expected to decrease in grassland-to-arable transition [89], nitrogen dynamics are highly temporal, with inputs and transformations occurring through complex interactions between soil, plants, microbes, air, and water.

Thus, these single snapshot measurements offer limited insight into nitrogen dynamics through key crop growth stages or how nitrogen cycled through the system. Instead, they offer insight only into the differences between the fertiliser treatments and management types at the time of sampling. No strong trends were observed, with the only noticeable difference being a slight reduction in both $N_{tot}$ and $N_{in}$ in the control plots under conventional management. Otherwise, values had high variance and exhibited substantial overlap between treatments, reinforcing the conclusion that neither fertiliser type nor management system had a strong or detectable effect on post-harvest soil nitrogen levels.

Total nitrogen ($N_{tot}$) comprised <1% of the oven-dry soil mass, with only around 0.1% of it present as inorganic nitrogen ($N_{in}$); the fraction readily available to plants. $N_{in}$ is chiefly comprised of two main forms, nitrate ($NO_3^-\_N$) and ammonium ($NH_4^+\_N$), which comprised approximately 75% and 25% of $N_{in}$ respectively.

$N_{tot}$ levels remained close to the baseline soil measurements, suggesting that total nitrogen stocks (principally organic N) were largely stable over the three-year period, regardless of fertilisation and management. This is somewhat surprising, given that humanure additions contained predominantly organic N (80–90%), in contrast to SF which was entirely inorganic. One might expect some accumulation of organic nitrogen in the soil from humanure applications; however, the absence of a clear increase in $N_{tot}$ suggests that this organic fraction was either effectively mineralised and taken up by crops or lost from the soil system, or was simply very small relative to the existing soil $N_{tot}$ pool.

## Phosphorus

Phosphorus (P) is a key macronutrient for plant growth, but many UK soils have elevated P levels due to historical over-application, leading to limited crop response to additional P and potential environmental risks [90]. As a result, current nutrient strategies aim to maintain, rather than increase, soil P by replacing only what is removed at harvest. The recommended level for cereals is P Index 2, or 16–25 mg/l plant-available P ($P_{av}$) [58,91].

Unfertilised and SF fertilised plots reported $P_{av}$ well below the Index 2 range for both management types, and slightly lower under regenerative management. Humanure plots under both management regimes showed significantly higher $P_{av}$ (conventional $M_{treatment}$ = 17.97 mg/kg; regenerative $M_{treatment}$ = 16.72 mg/kg), reaching the target P Index 2.

Without monitoring P dynamics over time, it is difficult to assess whether net accumulation (via direct inputs and mineralisation of organic P) or depletion (via plant uptake and losses from the system) took place, which could have been better identified with a greater number of measurements made throughout the experiment period. Whilst the recorded $P_{av}$ values fell within the target Index 2 range, the large increase over the other plots raises concerns about over-accumulation over time following repeat applications. These concerns are well established for sewage sludge biosolids [92–95], where it is noted that most biosolids applications apply more phosphate than is taken off in the following crop [96].

It should of course be noted that humanure differs greatly from biosolids in composition, so direct comparisons should be made cautiously, and a single-season snapshot cannot fully capture the complex, dynamic transformations of P in soil systems. Future research should monitor cumulative effects of repeated applications, and should determine the relative proportions of different forms of P within the compost material.

Interestingly, the applied $P_{av}$ from humanure was lower than from SF, yet the resulting soil $P_{av}$ increase was greater. This is likely due to the fact that a large percentage of the total P within organic material is in organic forms, and requires mineralisation to become plant-available [97–99]. Thus, the total P application was likely far higher in the humanure plots than the SF plots.

Estimates suggest that 50–80% of applied organic P from manures become plant available within a growing season [58,100,101], and these results indicate that some of this applied organic P was successfully mineralised, leading to the elevation of soil $P_{av}$. It's also possible that humanure influenced mineralisation of native soil organic P, as evidence suggests that soil organic matter increases abundance of phosphorus solubilising bacteria [102].

Finally, $P_{av}$ values were highly variable among the separate humanure plots, suggesting inconsistencies in mineralisation, uptake, or other environmental factors. This reinforces the need for a better understanding of P transformation dynamics during the composting process and within the soil system.

## Potassium

Potassium is an essential plant macronutrient, with recommended levels for arable crops in the UK at Index 2- (121–180 mg/kg) [103].

Humanure treatment elevated soil potassium ($K^+$) levels compared to the control and SF treatments, however most humanure values remained within the recommended Index 2- range. Under regenerative management, $K^+$ levels were higher for the control and SF plots than under conventional management, whereas humanure exhibited inconsistent trends. Humanure treatment was higher under regenerative management than conventional management in only one out of the three of the comparison plots.

Previous studies have shown that organic amendments, including manure and sewage sludge, can increase soil $K^+$ concentrations [104] and may act as a slow-release source of potassium, helping to maintain more stable soil levels compared to the short-term fluxes typically associated with inorganic K fertilisers [105].

There is also evidence that such amendments not only provide nutrients, but may also enhance plant uptake of nitrogen, phosphorus, and potassium [106]. This may contribute towards explaining the yield increases observed between humanure and the control.

As with phosphorus, it remains unclear whether ongoing humanure applications would lead to excessive potassium accumulation [107], or whether the inputs are balanced by plant uptake and losses.

Again, it is difficult to directly compare the inputs of potassium which are measured in different forms. Synthetic fertiliser contains mineral potassium, often as potash ($K_2O$). Organic materials will contain varying quantities of mineral potassium, exchangeable potassium (held on the surface of organic matter particles and readily available by exchange with other cations) and potassium stored within organic compounds which is less readily available [105,107,108].

Without a clear view of potassium additions, transformations and cycling, nor of the quantity of potassium taken up into the growing crops, a snapshot view of exchangeable K+ in the soil is of limited value [109]. More detailed analysis of potassium forms and fates would be needed to determine whether observed differences were due to differences in supply, transformation, losses or uptake.

## Worms

Earthworms are recognised as vital organisms in soil ecosystems, and their presence is often considered a good indicator of biological soil health [110,111]. They enhance soil aeration, which improves water drainage and root penetration to foster better conditions for plant growth [112]. Earthworms are also integral to the mixing and mechanical breakdown of organic matter (OM) into smaller fragments, enhancing contact surfaces for mineralising microorganisms [113] thus promoting nutrient cycling within the soil [114].

Since earthworm count is influenced by both seasonality and weather, comparisons between years are of limited use, and instead simultaneous measures taken on the same day provide the best insight into treatment differences.

The notably higher earthworm counts observed under regenerative management and under humanure addition suggests that these practices support favourable conditions. This is consistent with wider literature which finds that minimising physical disturbance increases earthworms, and also that organic material additions serve as a food source for worms [114–116]. Earthworms have also been found to be negatively impacted by pesticide application [117,118], which could contribute to the difference in worm abundance between the conventional and regenerative plots.

## Microbial respiration

Microbial respiration is a measure of the $CO_2$ released from soil as a result of the metabolism of organic matter by microorganisms, for energy and biomass production [119]. More broadly, it is a coarse and simple indicator of microbial

abundance and activity [120]. Soil microorganisms play a critical role in nutrient transformation processes and organic matter degradation, and many form important mutualistic associations with plant roots [121,122].

Under conventional management, humanure resulted in the highest microbial respiration rate of the three fertiliser treatments, indicating a positive effect. This is to be expected, as the addition of organic matter is positively correlated with microbial respiration rate [123–127]. A three year study by Iovieno et al. (2009) [123] found a positive relationship between food-waste compost and microbial activity (including respiration rate), and concluded the usefulness of repeated high-rate compost applications for soil microbial health.

Under regenerative management, the difference in respiration rate between humanure and the other fertiliser treatments was less pronounced, but the respiration rates for all three were elevated in comparison to the conventionally managed plots. This indicates a positive effect of regenerative management on microbial respiration rate. This is consistent with findings that tillage reduces overall microbial abundance [121,128] despite evidence of short initial bursts of $CO_2$ efflux immediately following tillage due to aeration and exposure of previously unavailable OM [129,130]. Additionally, the use of cover crops and diverse crop production regimes have been shown to increase abundance, activity and diversity of soil microbes [131,132]. Singh et al. (2023) [133] investigated the effect of multiple regenerative practices on bacterial community structure over a 3 and 5-year experiment and found they were all effective in improving bacterial community structure and overall soil health.

Conversion of grassland to arable land is expected to broadly decrease microbial diversity [134,135], however effects on abundance and composition remain weakly understood [21,136]. It is expected that the microbial abundance of the soil declined over the study period, but the method used is not appropriate for meaningful comparison between the start and end of the trial, due to its temperature and moisture sensitivity. A more detailed analysis of community structure at numerous timesteps would be needed to determine the effects arising from grassland-to-arable transition. Instead, these results offer a broad view of the impacts of different land management and fertiliser additions.

### Fungi

Fungi are important indicators of biological soil health due to their ubiquity, diversity and sensitivity to environmental change [137]. This diverse kingdom plays key roles in organic matter decomposition, nutrient cycling, plant pathogen suppression, and carbon sequestration [138,139].

It is expected that conversion of grassland to more intensively managed arable land will decrease fungal diversity and abundance due to increased disturbance [140]. This is supported by the fundings of this study, which found that regenerative management practices showed slightly higher fungal abundance than conventional practices. This aligns with existing literature demonstrating the detrimental effects of tillage on fungal communities [128] and the beneficial impact of cover cropping [141].

That said, no baseline data was available for comparison, and the data should be interpreted with caution. Fungal biomass measurements exhibited considerable variability, even among triplicates taken from the same sample bag. This high within-treatment variation limits the reliability of the results and raises concerns about the suitability of the method for this type of analysis. These findings highlight a broader challenge in soil biology: the difficulty of obtaining accurate and reproducible estimates of fungal biomass using simple and cost-effective approaches. To improve confidence in future results, either alternative, more robust methods should be adopted, or the number of true replicates should be substantially increased.

### Composition and use of humanure

The composition of the three humanure batches varied considerably (S2 Table), as expected for materials produced under different conditions [9,142]. While total nitrogen ($N_{tot}$) was comparable across batches, quantities and forms of inorganic nitrogen ($N_{in}$) differed significantly, reflecting the complex and variable nature of N

transformation during storage. Despite this, C:N ratios (13.59–16.07) aligned closely with the expected range for mature composts [143,144].

High within-batch variability further illustrates the heterogeneity of homemade humanure and the challenge of achieving uniform nutrient content. This complicates accurate dosing for scientific research and poses a barrier to potential commercial applications, where standardisation may be required. Continued reporting of humanure composition from different systems is therefore vital to help establish reliable reference ranges.

In this study, a matched $N_{tot}$ approach was taken between humanure and synthetic fertiliser (SF). This is a common approach [9], and the quantities of humanure that would have been required to match $N_{in}$ would have been infeasible. However, organic N mineralises slowly, and estimates vary for the quantity which becomes plant-available in each growing season. A review by Rigby et al. (2016) [145] suggests this value falls between 2–24.5% for composted biosolids, whilst research on animal manures suggests a range of −2% to 48%, with plant-available nitrogen (PAN) generally showing an inverse relationship with C:N ratio [146]. A value of 10% PAN is sometimes adopted as a guideline figure [145], with an understanding that smaller proportions continue to be mineralised in subsequent years, contributing to a long-term residual effect [147].

The slow mineralisation of organic N introduces uncertainty in nutrient supply, both for scientific comparison and for practical use. Future studies might better align fertiliser equivalency by matching PAN rather than $N_{tot,}$ though this approach requires careful year-on-year adjustment to account for ongoing mineralisation.

## Conclusion

This study explored the impacts of applying humanure as a fertiliser during grassland-to-arable transition, under both conventional and regenerative land management approaches. While findings are preliminary due to the small sample sizes, they provide useful insights into how humanure interacts with different management practices, and where it may best fit into future agricultural strategies.

In conventional systems, humanure showed potential as a direct replacement for synthetic fertiliser. Although crop performance under humanure was slightly lower overall, a high degree of overlap in the data suggests that these differences may not be statistically significant. This reduction in yield was likely driven by a lower number of grains per head, potentially linked to limited nitrogen availability early in the growing season due to the slower nutrient release profile of humanure. However, humanure treatments led to notably higher biological indicators—specifically microbial respiration and earthworm abundance—compared to both synthetic fertiliser treatment and the control. Physical soil characteristics showed little meaningful variation across treatments, while soil organic matter, available phosphorus, and potassium were all elevated under humanure application. Total and inorganic soil nitrogen were similar under humanure and SF application, and higher than the control.

The observed soil health benefits under humanure fertilisation may take longer than three years to confer any benefit to crop production, with previous studies indicating that it may take longer than three years for organic production to 'catch up' to conventional production [148]. More long-term studies are sorely needed within this area of research to fully model this transition time. Practical challenges also remain for direct substitution at farm scale, including the large quantities of humanure required, logistical issues related to transport and application, and the difficulty in delivering a standardised product, both for informed nutrient management and to guarantee product safety.

Under the regenerative system, humanure performed particularly well. It was hypothesised that the enhanced biological activity associated with regenerative practices would better facilitate nutrient cycling from the organic inputs, and the results support this. Yields under regenerative humanure treatment exceeded both the regenerative control and the conventional humanure and synthetic fertiliser treatments, due mainly to increased grain number and a reduction in damage and disease-related losses. Regenerative-humanure treatment also saw the highest SOM content, microbial respiration rates, and earthworm counts across all plots, indicating a positive interaction effect.

This is a notable finding, especially after the review by Allen et al. (2023) [9] highlighted a lack of reporting on biological and physical soil health parameters in studies using faecal-based amendments. By assessing these indicators, this study begins to fill that gap and suggests that the combination of regenerative management and organic waste amendments like humanure may provide synergistic benefits for both soil and crop outcomes.

Moreover, many previous studies have assessed humanure within single year, isolated frameworks; this study's design allowed for a side-by-side comparison across two contrasting systems over a three-year period of grassland-to-arable transition, revealing for the first time clear differences in fertiliser performance depending on the wider management context.

Additionally, regenerative practices consistently outperformed conventional management across all fertiliser treatments for most crop performance indicators, as well as SOM, water-stable aggregates, and biological soil health. This suggests that regenerative management alone may improve soil function and productivity, challenging the idea that conventional management should be the default benchmark. This insight has major implications for the way we model and design waste reuse systems: rather than focusing solely on input substitution, we may need to rethink the management context entirely to best match resources with appropriate land management contexts.

These findings clearly suggest that the lower disruption of regenerative practices may facilitate better grassland-to-arable transition than conventional practices, by minimising the negative impacts on soil health, particularly soil organic matter and biological indicators.

Macronutrient dynamics were a key point of discussion; under both management regimes, humanure application resulted in soil available phosphorus ($P_{av}$) and potassium (K+) levels within agronomic target ranges, and significantly higher than both the controls and the synthetic fertiliser treatments. The effect of management type alone was comparatively minor. In the absence of robust, repeated measures over time it was not possible to infer whether these values were declining from elevated starting conditions, indicative of differential rates of depletion, or increasing due to high application within humanure which could risk long-term over-accumulation. Further research is warranted.

Finally, it is important to recognise the limitations of the study. Sample sizes were small—a constraint common to plot-scale agricultural experiments—and this restricted the statistical power of the results. Observed trends should thus be considered exploratory, rather than definitive. Nonetheless, the observed patterns offer promising directions for future research.

Taken together, these findings indicate that i) regenerative practices may offer a realistic pathway for successful productivity during grassland-to-arable transition, whilst retaining soil health, and ii) humanure holds promise as a fertiliser input, particularly within regenerative systems where biological activity can support decomposition and nutrient mineralisation of the material.

This supports existing findings of positive fertility and soil health benefits of sewage sludge biosolids, compared with synthetic fertiliser [34,149,150], and holds particular promise for grassland-to-arable transition in urban and peri-urban areas which are better co-located with high supply of fecal biomass [21,22]. Both the integration of recycled organic waste streams into agricultural production and the transition towards more regenerative production systems could help to reduce dependence on synthetic fertilisers, whilst also safeguarding soil health and closing nutrient loops. With further practical and safety considerations, these approaches are well positioned to support more sustainable agricultural expansion, without compromising productivity.

## Supporting information

**S1 Table. Detailed plot management diary.**
(DOCX)

**S1 Appendix. Humanure batch collection and sampling.**
(DOCX)

**S2 Table. Chemical composition of the three humanure batches used in each year of the experiment.**
(DOCX)

**S2 Appendix. Detailed lab methods.**
(DOCX)

**S1 Fig. Barley head damage and disease examples.**
(DOCX)

**S1 Data. Full raw 2024 data.**
(CSV)

**S2 Fig. Spring barley establishment.**
(DOCX)

**S3 Fig. Spring barley tiller number per plant.**
(DOCX)

**S4 Fig. Spring barley stem height.**
(DOCX)

**S5 Fig. Spring barley grain number.**
(DOCX)

**S6 Fig. Spring barley 1000 grain mass.**
(DOCX)

**S7 Fig. Spring barley dry head biomass.**
(DOCX)

**S8 Fig. Soil dry bulk density after three years of cropping.**
(DOCX)

**S9 Fig. Soil nitrate nitrogen content after three years of cropping.**
(DOCX)

**S10 Fig. Soil ammonium nitrogen content after three years of cropping.**
(DOCX)

## Author contributions

**Conceptualization:** Katie Allen, Ruth Wade, Barbara Evans.

**Data curation:** Katie Allen.

**Formal analysis:** Katie Allen.

**Investigation:** Katie Allen.

**Methodology:** Katie Allen.

**Project administration:** Katie Allen, Ruth Wade.

**Supervision:** Effie Papargyropoulou, Ruth Wade, Barbara Evans.

**Visualization:** Katie Allen.

**Writing – original draft:** Katie Allen.

**Writing – review & editing:** Katie Allen, Effie Papargyropoulou, Ruth Wade, Barbara Evans.

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
