## [Decision Letter · Decision Letter 0]

24 Jul 2025

Dear Dr. Allen,

Thank you for submitting your manuscript to PLOS ONE. After careful consideration, we feel that it has merit but does not fully meet PLOS ONE’s publication criteria as it currently stands. Therefore, we invite you to submit a revised version of the manuscript that addresses the points raised during the review process.

We look forward to receiving your revised manuscript.

Kind regards,

Paulo H. Pagliari

Academic Editor

PLOS ONE

Journal Requirements:

3. In the online submission form, you indicated that [Data available upon request.].

Additional Editor Comments:

Please see the reviewer's comments for how to improve the manuscript.

Reviewers' comments:

Reviewer's Responses to Questions

**Comments to the Author**

1. Is the manuscript technically sound, and do the data support the conclusions?

Reviewer #1: Yes

2. Has the statistical analysis been performed appropriately and rigorously?

Reviewer #1: Yes

3. Have the authors made all data underlying the findings in their manuscript fully available?

Reviewer #1: No

4. Is the manuscript presented in an intelligible fashion and written in standard English?

Reviewer #1: Yes

Reviewer #1: See my comments embedded in the attached reviewed file.

Here, for example, is my last comment:

This is truly a ground-breaking study (pun intended). Hats off to the researchers! FYI, the Glastonbury music festival utilizes what are probably “dry toilets” (as opposed to compost toilets) where a LOT of toilet material is collected and presumably composted off-site. From what I can tell, these toilets are being well-managed and could be a great source of material for future research.

I don’t see any major problems with your research, other than the minor issues I have mentioned in my comments. Better sources of humanure would likely make a major difference in outcomes.

Get your hands on the Humanure Handbook 4th edition (2019), as well as the Compost Toilet Handbook (2021). They can be downloaded as PDFs online. I can provide free PDF copies to researchers. Just email me a request (joe@josephjenkins.com).

Joe Jenkins

**Do you want your identity to be public for this peer review?** For information about this choice, including consent withdrawal, please see our Privacy Policy

Reviewer #1: **Yes:** Joseph Jenkins

---

## [Author Response · Author response to Decision Letter 1]

4 Sep 2025

Response to Reviewers 04/09/2025

Response to Academic Editor

Thank you for your swift response. I have made the requested changes and reuploaded the tracked changes and clean versions of the manuscript:

- I have included a statement which indicates additional permits were not required as the study was conducted on a university-owned research site.

- I have uploaded all the raw field data as an additional file.

- I have reformatted the manuscript headings and supporting information files to match the style guides.

Reviewer 1 Comments and Responses:

Abstract: This needs to be defined in some detail. The term is used throughout the paper, but it is not clear what it means.

> The term first appears in line 83, and the next paragraph (Line 84) explains the need to include specificity when using these kinds of terms which have multiple different interpretations.

> The section beginning at Line 120 provides more detail about those features which are commonly considered under the ‘regenerative’ umbrella, and explains those features which were included within this experiment.

> A full list of these practices undertaken on the regenerative plots is given in the Methods section in Table 1.

Line 12: "Humanure" is human excrement (primariy fecal material and urine) recycled for agricultural purposes. It is recommended that the "toilet material" be composted prior to agricultural use. There is no evidence presented here that the humanure was composted.

Composting, by definition, requires the production of internal biological heat in the organic material being composted, as described in the Humanure Handbook 4th edition (2019). The 1st edition of the Humanure Handbook (1994) was used as a reference in this study.

The use of the word "humanure" in the title to this paper is correct. The humanure used in this study consisted of the aged contents of "dry toilets.," which consisted of humanure, along with whatever other contents were deposited in the toilets.

If the toilet contents had been composted, there should be data regarding the organic materials that were utilized (humanure and what else), plus time and temperature factors. "Sawdust" was added, but was it sawdust, wood shavings, or wood chips? Was urine separated from the toilet contents, as is often the case with dry toilets?

Separation of urine and inclusion of large carbon chunks (wood chips), will drastically alter the nutrient composition of the organic mass. These are important factors, and this sort of information, if included, would improve this research paper.

To rectify this, I would suggest changing the definition of humanure as stated from "composted human feces" to something like "human excrement recycled for agricultural purposes."

I expect that you would find that correctly composted human toilet material, including the urine, when mixed with other organic discards in the composting process. (kitchen food, agricultural byproducts, and food/beverage industry discards), will provide better agricultural outcomes. Not that the outcomes were bad here, but just saying.

> Thank you for this insightful comment; this nuance has been a tricky point throughout my research, especially when considering the differences communicating with agricultural experts, sanitation experts, and non-expert audiences. It is something I have discussed at great length in my thesis, and challenged the idea that humanure and composting are always synonymous, and suggest some alternative and nuanced definitions.

But, that’s away from the present point. I accept your comment here, that for this instance in this paper the phrasing is indeed wrong. As such I have amended it to your suggested wording, and have increased the amount of detail given for each humanure batch used in the experiment.

Line 192: What kind(s) of sawdust? From trees or lumber? Fresh sawdust or aged? Actual sawdust, or woodchips or wood shavings (see Humanure Handbook 4th edition (2019).

Thank you for raising the need for more specificity here. I have updated this line to explain that the batches differed, and have also included more details about each batch in the Supporting Information 2 file. Since these materials were arriving to me second-hand, where information about the contents is unknown, this is now made more clear.

Line 361: Typo. should be "do"?

> Well spotted, thank you! Fixed.

Line 379: typo

> Well spotted, thank you! Fixed.

Line 582: Are these words necessary?

> A valid question. They were included in attempts to move away from yield being perceived as the only metric of success, as this paper also considers the impacts on improving soil health. The sentence has been reworded for clarity.

Line 625: typo

> Well spotted, thank you! Fixed.

Line 629: typo

> Well spotted, thank you! Fixed.

Line 672: should be two words

> Well spotted, thank you! Fixed.

Line 854: This is truly a ground-breaking study (pun intended). Hats off to the researchers!

FYI, the Glastonbury music festival utilizes what are probably “dry toilets” (as opposed to compost toilets) where a LOT of toilet material is collected and presumably composted off-site. From what I can tell, these toilets are being well-managed and could be a great source of material for future research.

I don’t see any major problems with your research, other than the minor issues I have mentioned in my comments. Better sources of humanure would likely make a major difference in outcomes.

Get your hands on the Humanure Handbook 4th edition (2019), as well as the Compost Toilet Handbook (2021). They can be downloaded as PDFs online. I can provide free PDF copies to researchers. Just email me a request (joe@josephjenkins.com).

Joe Jenkins

> Thank you for these kind words, and for taking the time to review my research. I really appreciate it!

---

## [Decision Letter · Decision Letter 1]

29 Oct 2025

Dear Dr. Allen,

Thank you for submitting your manuscript to PLOS ONE. After careful consideration, we feel that it has merit but does not fully meet PLOS ONE’s publication criteria as it currently stands. Therefore, we invite you to submit a revised version of the manuscript that addresses the points raised during the review process.

We look forward to receiving your revised manuscript.

Kind regards,

Paulo H. Pagliari

Academic Editor

PLOS ONE

Journal Requirements:

Additional Editor Comments:

Katie, please go ahead and make the changes we spoke about and please make sure you use work track changes so that I can see what was changed easily and upload the revised version.

Reviewers' comments:

Reviewer's Responses to Questions

**Comments to the Author**

Reviewer #1: (No Response)

2. Is the manuscript technically sound, and do the data support the conclusions?

Reviewer #1: Yes

3. Has the statistical analysis been performed appropriately and rigorously?

Reviewer #1: N/A

4. Have the authors made all data underlying the findings in their manuscript fully available?

Reviewer #1: No

5. Is the manuscript presented in an intelligible fashion and written in standard English?

Reviewer #1: Yes

Reviewer #1: I think I would have first blended all of the humanure batches together thoroughly, then used the mix for this research, as it could have eliminated some of the discrepancies in the results. In short, I think this paper will likely inspire more researchers to study this subject matter. There have been many great opportunities to do so, but nobody to do the research. We composted hundreds of tons of humanure with sugarcane bagasse and food scraps in Haiti, only to have the project abandoned due to security problems when it was time to use the finished compost agriculturally. In the end, local farmers came and took the compost and used it for agricultural purposes, but nobody documented anything (see, for example https://youtu.be/VY5K2Jn7Om0?si=38vI5QET-RcMgHFN). I should add that this situation is occurring on several continents where large-scale humanure composting is taking place with little or no follow-up agricultural research. See the Compost Toilet Handbook for more information.] Good job with this research!

**Do you want your identity to be public for this peer review?** For information about this choice, including consent withdrawal, please see our Privacy Policy

Reviewer #1: **Yes:** JOSEPH JENKINS

---

## [Author Response · Author response to Decision Letter 2]

19 Dec 2025

These comments to reviewers were addressed in the previous resubmission. This latest resubmission includes some additional information and changes which were suggested as a result of my PhD viva corrections.

---

## [Editor Report · Decision Letter 2]

8 Jan 2026

The use of humanure for cereal production under conventional and regenerative farming models - findings from a three-year grassland-to-arable transition

PONE-D-25-29073R2

Dear Dr. Allen,

We’re pleased to inform you that your manuscript has been judged scientifically suitable for publication and will be formally accepted for publication once it meets all outstanding technical requirements.

Kind regards,

Paulo H. Pagliari

Academic Editor

PLOS One
---

## [Editor Report · Acceptance letter]

PONE-D-25-29073R2

PLOS One

Dear Dr. Allen,

I'm pleased to inform you that your manuscript has been deemed suitable for publication in PLOS One. Congratulations! Your manuscript is now being handed over to our production team.

Kind regards,

on behalf of

Dr. Paulo H. Pagliari

Academic Editor

PLOS One